# Targeting SWI/SNF ATPases reduces neuroblastoma cell plasticity

Man Xu ⓘ, Jason J Hong ⓘ, Xiyuan Zhang, Ming Sun, Xingyu Liu, Jeeyoun Kang, Hannah Stack ⓘ, Wendy Fang, Haiyan Lei, Xavier Lacoste, Reona Okada, Raina Jung, Rosa Nguyen, Jack F Shern, Carol J Thiele ⓘ ✉ & Zhihui Liu ⓘ ✉

## Abstract

Tumor cell heterogeneity defines therapy responsiveness in neuroblastoma (NB), a cancer derived from neural crest cells. NB consists of two primary subtypes: adrenergic and mesenchymal. Adrenergic traits predominate in NB tumors, while mesenchymal features becomes enriched post-chemotherapy or after relapse. The interconversion between these subtypes contributes to NB lineage plasticity, but the underlying mechanisms driving this phenotypic switching remain unclear. Here, we demonstrate that SWI/SNF chromatin remodeling complex ATPases are essential in establishing an mesenchymal gene-permissive chromatin state in adrenergic-type NB, facilitating lineage plasticity. Targeting SWI/SNF ATPases with SMARCA2/4 dual degraders effectively inhibits NB cell proliferation, invasion, and notably, cellular plasticity, thereby preventing chemotherapy resistance. Mechanistically, depletion of SWI/SNF ATPases compacts cis-regulatory elements, diminishes enhancer activity, and displaces core transcription factors (MYCN, HAND2, PHOX2B, and GATA3) from DNA, thereby suppressing transcriptional programs associated with plasticity. These findings underscore the pivotal role of SWI/SNF ATPases in driving intrinsic plasticity and therapy resistance in neuroblastoma, highlighting an epigenetic target for combinational treatments in this cancer.

**Keywords** SWI/SNF Complexes; Core Transcription Factors; Neuroblastoma; Epigenetic Plasticity; Cancer Cell Plasticity
**Subject Categories** Cancer; Chromatin, Transcription & Genomics

## Introduction

Neuroblastoma (NB) is the most common extracranial solid tumor in children, and although treatments have improved, the 5-year survival rate for high-risk NB patients remains under 50% as these patients are refractory to current therapies or relapse (Cheung and Dyer, 2013; Huang and Weiss, 2013; Irwin et al, 2021; Qiu and Matthay, 2022). *MYCN* amplification occurs in over 20% of NB patients, with MYCN playing a pivotal role as an oncogenic driver that significantly contributes to uncontrolled NB growth and metastasis (Huang and Weiss, 2013; Liu et al, 2020a; Otte et al, 2020; Teitz et al, 2013; Weiss et al, 1997; Zhu et al, 2017). NB has been classified into two major subtypes: undifferentiated mesenchymal (MES), and the predominant subtype of lineage-committed adrenergic (ADRN) NB, which are defined by core transcriptional regulatory circuitry (CRC) transcription factors (TFs) (Boeva et al, 2017; Durbin et al, 2018; van Groningen et al, 2017). Importantly, the interchangeability between these subtypes introduces intra-tumoral heterogeneity and endows NB with high plasticity (Thirant et al, 2023; van Groningen et al, 2017). Tumor cell plasticity drives their transition towards a phenotypic state that is no longer reliant on the drug-targeted pathway, leading to drug resistance (Boumahdi and de Sauvage, 2020; Davies et al, 2023; Perez-Gonzalez et al, 2023). In NB, MES-type cells become enriched following chemotherapy and in relapsed tumors, exhibiting increased resistance to chemotherapeutic drugs, which potentially contributes to the relapse and cancer progression (Boeva et al, 2017; Durbin et al, 2018; Gartlgruber et al, 2021; Gautier et al, 2021; Manas et al, 2022; Ponzoni et al, 2022; Thirant et al, 2023; van Groningen et al, 2017). MYCN forms a network with the CRC TFs HAND2, PHOX2B, GATA3, and other TFs to establish the ADRN phenotype, while PRRX1, NOTCH3, and others determine the MES phenotype (Boeva et al, 2017; De Wyn et al, 2021; Decaesteker et al, 2018; Durbin et al, 2018; van Groningen et al, 2017; Wang et al, 2019; Xu et al, 2023). However, the epigenetic mechanisms governing NB plasticity have not been fully characterized.

The structure of chromatin has a significant impact on cellular identity and transitions between different states. Genetic, environmental, or metabolic insults can lead to overly restrictive or overly permissive epigenetic landscapes, contributing to cancer pathogenesis (Flavahan et al, 2017; Vicente-Duenas et al, 2018). Restrictive chromatin states may prevent the proper activation of tumor suppressor programs or block differentiation, whereas permissive states can allow for stochastic activation of oncogenes or non-physiologic cell fate transitions (Flavahan et al, 2017). Similar epigenetic mechanisms may be involved in regulating cancer cell plasticity. In eukaryotic cells, the presence of nucleosomes typically creates a barrier for the DNA binding of TFs involved in the regulation of specific genes (Clapier and Cairns, 2009; Li et al, 2007). The mammalian switch/sucrose non-fermenting (mSWI/SNF) family of chromatin-remodeling complexes utilize either ATPase SMARCA4 or

Pediatric Oncology Branch, Center for Cancer Research, National Cancer Institute, Bethesda, MD, USA. ✉E-mail: thielec@mail.nih.gov; liuzhihu@mail.nih.gov

SMARCA2 as mutually exclusive catalytic subunits to remodel chromatin through processes such as nucleosome sliding and eviction. By creating nucleosome-depleted regions, SWI/SNF ATPases facilitate access of TFs to chromatin, thereby controlling pluripotency and cell-fate specification (Centore et al, 2020; Clapier et al, 2017; Hargreaves, 2021; Kadoch and Crabtree, 2015; Mittal and Roberts, 2020; Xiao et al, 2022). These findings suggest that SWI/SNF ATPases may potentially play a role in regulating cancer cell plasticity.

Cancer genome-sequencing studies have revealed mutations in genes encoding subunits of SWI/SNF complexes in over 20% of all cancers (Centore et al, 2020; Kadoch and Crabtree, 2015; Mittal and Roberts, 2020). SMARCA4 and SMARCA2 have emerged as oncogenic dependencies in certain cancer types, with SMARCA4 specifically being implicated in cancer metastasis (Farnaby et al, 2019; Huang et al, 2018; Kofink et al, 2022; Papillon et al, 2018; Saladi et al, 2010; Xiao et al, 2022). Notably, the targeting of SWI/SNF ATPases using the PROTAC degraders, ACBI1 and AU-15330, or a selective catalytic inhibitor BRM014, has demonstrated suppression of tumor growth across multiple cancer types (Farnaby et al, 2019; Kofink et al, 2022; Papillon et al, 2018; Xiao et al, 2022), underscoring the potential of SWI/SNF ATPases as therapeutic targets in cancers. In the context of NB, genetic alterations affecting subunits of the SWI/SNF complex, such as *SMARCA4, ADRID1A,* and *ADRID1B*, have been identified in over 10% of NB patients (Bellini et al, 2019; Sausen et al, 2013; Witkowski et al, 2023). These alterations often lead to loss of function effects due to large deletions, gene truncations, and loss of heterozygosity, implying their role as classic tumor suppressors (Bellini et al, 2019; Sausen et al, 2013; Witkowski et al, 2023). However, studies investigating the biological function of SWI/SNF complexes in NB have yielded conflicting results. ARID1A has been recognized as a tumor suppressor (Li et al, 2017; Shi et al, 2020), while disrupting the canonical SWI/SNF complex structure through the silencing of both *ARID1A* and *ARID1B* has been found to inhibit the growth and metastasis of NB (Jimenez et al, 2022). Moreover, there are conflicting reports on the role of SWI/SNF ATPases in regulating NB cell growth (Jimenez et al, 2022; Jubierre et al, 2016). Consequently, there is a need for a more comprehensive molecular understanding of the role of SWI/SNF chromatin remodeling activity in regulating NB tumor biology.

In this study, we discovered that targeting SWI/SNF ATPases not only inhibits NB cell proliferation, migration, invasion, and plasticity but also sensitizes NB cells to chemotherapeutic drug treatment. Mechanistically, the depletion of SWI/SNF ATPases induces chromatin compaction, diminishes enhancer activity, and disrupts the binding of MYCN, HAND2, PHOX2B and GATA3 to DNA. This disruption, in turn, suppresses downstream oncogenic and invasive transcriptional programs, including those associated with mesenchymal and adrenergic gene expression that contribute to NB plasticity. These findings underscore the potential of SWI/SNF ATPases as promising targets for precision therapy in high-risk NB cases that rely on core TFs. Furthermore, our results suggest that SWI/SNF ATPases may play a role in mediating plasticity in other types of cancer as well.

# Results

## Targeting SWI/SNF ATPases inhibits NB growth

To elucidate the role of SWI/SNF ATPases in NB, we first evaluated *SMARCA2/4* expression in the context of normal tissues and cancers. Analysis of the GENT2 gene expression database (gent2.appex.kr) encompassing over 68,000 samples from 72 different tissues (Park et al, 2019) revealed significantly higher levels of *SMARCA4* mRNA in all-cancers compared to all-normal tissues in two different datasets (Appendix Fig. S1A), while *SMARCA2* mRNA levels were significantly lower in all-cancers compared to all-normal tissues in two different datasets (Appendix Fig. S1B). By using the depmap online platform (depmap.org), we observed that NB exhibited significantly higher *SMARCA4* mRNA levels compared to most other cancer types (Fig. 1A, left panel). Moreover, analysis of genome-wide CRISPR screens for cancer genetic vulnerability (depmap.org) demonstrated the selective essentiality of *SMARCA4* in NB cell lines, distinguishing it from most other cancers (Fig. 1A, right panel). In contrast, analysis of the depmap datasets revealed lower *SMARCA2* mRNA levels in NB cell lines compared to other cancer cell lines, and *SMARCA2* was not identified as essential to NB cell growth (Fig. 1B). Evaluation of publicly available data indicates *SMARCA4* mRNA levels are significantly higher compared to *SMARCA2* mRNA levels in NB cell lines (depmap.org) as well as in NB patients' tumors (r2.amc.nl) (Appendix Fig. S1C). Elevated levels of SMARCA4 protein and lower levels of SMARCA2 protein were detected in NB cell lines, compared to the 'non-tumorigenic', human retinal pigmented epithelial cell line ARPE-19, which is commonly used as a non-transformed control in NB studies (Appendix Fig. S1D). To investigate the correlation of *SMARCA2/4* mRNA expression with NB patient survival, we analyzed three clinically annotated NB cohorts (r2.amc.nl). We found that high expression of *SMARCA4* correlated with poor overall survival (Appendix Fig. S1E), while there was no significant correlation between *SMARCA2* expression and overall survival in NB (Appendix Fig. S1F). This was consistent with a previous report (Jubierre et al, 2016).

By exploring the predictability function in DepMap, we observed that low levels of *SMARCA2* are positively correlated with *SMARCA4* CRISPR knockout effects across all cancer cell lines (Appendix Fig. S1G, left panel). In NB cell lines, this correlation is not as significant, but there is a trend of positive correlation (Appendix Fig. S1G, right panel). Additionally, low *SMARCA4* expression positively correlates with *SMARCA2* CRISPR knockout effects when considering all cancer cell lines (Appendix Fig. S1H, left panel). However, this positive correlation is not observed when restricting our analysis to NB cell lines (Appendix Fig. S1H, right panel). Nevertheless, NB cell lines with lower *SMARCA2* expression appear to be more sensitive to the loss of *SMARCA4*, suggesting that targeting both SMARCA2 and SMARCA4 could be a promising approach for treating NB.

To assess the biological function of SWI/SNF ATPases in NB cells, we knocked down *SMARCA4* or *SMARCA2* using siRNAs and observed a significant decrease in cell proliferation after silencing of *SMARCA4* but not *SMARCA2* (Appendix Fig. S1I–K). This is consistent with the genome-wide CRISPR screen data showing that NB cells are dependent on *SMARCA4* but not *SMARCA2* for cell growth (Fig. 1A,B). Given the lack of selective SMARCA2 or SMARCA4 inhibitors, we treated a panel of ADRN-type NB cell lines with the SMARCA2/4 dual PROTAC degraders ACBI1 or AU-15330. Western blot analysis demonstrated that treatment of NB cells with ACBI1 or AU-15330 led to a time- and dose-dependent reduction in SMARCA4 and SMARCA2 protein levels (Fig. 1C,D; Appendix Fig. S1L,M). Moreover, SMARCA2/4 protein

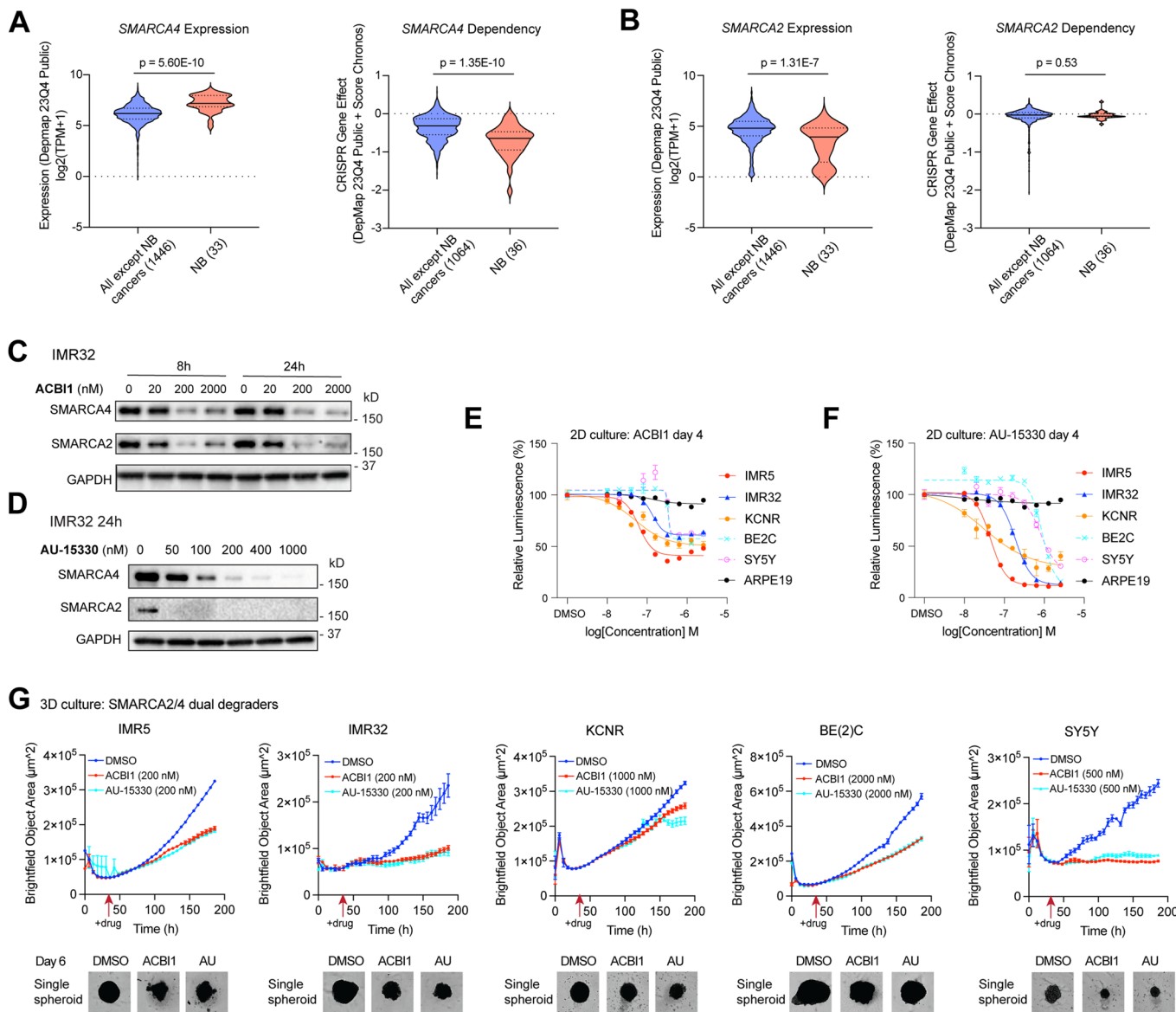

**Figure 1. Targeting SWI/SNF ATPases inhibits NB growth.**

(A, B) Comparison of *SMARCA4* and *SMARCA2* mRNA levels between NB (*n* = 33) and other cancer cell lines (*n* = 1446) (queried from depmap.org) (left panels). Determine *SMARCA4* and *SMARCA2* dependencies in NB (*n* = 36) and other cancer cell lines (*n* = 1064) by analyzing data from genome-wide cancer genetic vulnerability CRISPR screens on DepMap (right panels). (C, D) Protein levels of SMARCA2 and SMARCA4 in IMR32 cells were detected by western blot assay after treatment with different doses of PROTAC degraders ACBI1 and AU-15330. GAPDH is used as a loading control. (E, F) CellTiter-Glo assay measuring the of ACBI1 or AU-15330 treatment on cell growth (*n* = 3/group; Error bars indicate SEM). The cell viabilities of DMSO-treated cells are set to 100%, and IC-50 curves are generated using the GraphPad Prism software. Data are representative of two independent experiments. (G) Impact of ACBI1 or AU-15330 treatment on NB cell single spheroid growth in liquid 3D culture evaluated through Incucyte Spheroid Growth Assays (*n* = 4 or 6/group; Error bars indicate SEM). Data are representative of three independent experiments. Data information: In panels (A, B), data are presented as violin plots, where the middle solid lines indicate medians and the dash lines represent the 25th and 75th percentiles. The number of samples is shown on the graph. Statistical differences were calculated using a two-sided unpaired Student's *t*-test. Source data are available online for this figure.

levels remain low even at day 5 without replenishing the drug supply (Appendix Fig. S1N,O). This suggests that the half-life of ACBI1 and AU-15330 is greater than 5 days under two-dimensional (2D) cell culture conditions. Pharmacological targeting of SWI/SNF ATPases with either ACBI1 or AU-15330 resulted in a dose-dependent reduction in cell proliferation of NB cell lines (Fig. 1E,F). Similar to what has been seen in other studies in which

the growth of several non-tumorigenic cell lines, including HEK 293, is not affected by SMARCA2/4 degrader AU-15330 (Xiao et al, 2022), we found the proliferation of the non-tumorigenic ARPE-19 cell line, was not affected by the treatment of SMARCA2/4 degraders (Fig. 1E,F), even though low doses of the degraders efficiently depleted SMARCA2/4 proteins in ARPE-19 cells (Appendix Fig. S1L,M).

Both western blot results and cell proliferation assays indicate varying sensitivities to SMARCA2/4 degrader among different NB cell lines. For instance, BE(2)C cells exhibit the highest resistance to drug treatment in two-dimensional cell proliferation assays (Fig. 1E,F), necessitating higher doses to achieve a greater than 80% degradation of SMARCA2/4 (Appendix Fig. S1L,M). The distinct responses of NB cell lines to ACBI1 and AU-15330 treatment may arise from genetic differences among NB cell lines, for instance, p53 mutation occurs in BE(2)C but not IMR5 or IMR32 (as detailed in the Materials and Methods section). Additionally, differences in the proteasome machinery in NB cells may affect their degradation ability as well as other factors affecting drug penetration and efflux regulators (Luo et al, 2022; Ottis et al, 2019; Zhou et al, 2023). Consequently, we selected optimal doses of ACBI1 or AU-15330 that can decrease SMARCA2/4 levels by more than 80% to treat NB cells in subsequent studies.

To explore the physiological relevance of SWI/SNF ATPases targeting, we employed three-dimensional (3D) spheroids of NB cells, which mimic in vivo tumor characteristics (Nath and Devi, 2016; Weiswald et al, 2015). We found that the treatment of single spheroids with ACBI1 or AU-15330 also led to decreased cell proliferation in NB cell lines (Fig. 1G; Appendix Fig. S1P).

Genetic knockdown of *SMARCA4* in NB cells leads to apoptotic cell death (Jubierre et al, 2016). To investigate the impact of ACBI1 and AU-15330 on NB cell apoptosis, we treated IMR5, IMR32, and BE(2)C cell lines with varying doses of ACBI1 or AU-15330. We performed an apoptosis assay by incubating the cells with Incucyte Caspase-3/7 green dye. Representative images of IMR5 cells showed that ~50% of cells treated with 200 nM ACBI or AU-15330 for 4 days exhibited green fluorescence (Appendix Fig. S1Q, left panel). Additionally, cell image analysis revealed that various doses of ACBI1 or AU-15330 treatment led to a significant increase in cell apoptosis (Appendix Fig. S1Q, right panel). However, less than 10% of apoptotic cells were detected in IMR32 and BE(2)C cells after 4 days of ACBI or AU-15330 treatment (Appendix Fig. S1R,S). These findings indicate that there is variable induction of apoptosis after the depletion of SWI/SNF ATPases using PROTAC degraders in different NB cell lines.

## Targeting SWI/SNF ATPases leads to the compaction of *cis*-regulatory elements bound by core TFs

To examine the impact of SWI/SNF ATPases depletion on chromatin accessibility, we performed an assay for transposase-accessible chromatin followed by sequencing (ATAC-seq). In IMR32 cells, out of the 129,452 ATAC-seq peaks, ACBI1 treatment (24 h) resulted in decreased chromatin accessibility (>2.5-fold changes) in 22,070 peaks (17.05%), with around 1044 peaks (0.81%) showing increased chromatin accessibility (Fig. 2A, left panel). Additionally, 40.66% showed subtle changes (1.25–2.5-fold), while 41.49% remained unaffected by ACBI1 treatment (within 1.25-fold changes). In BE(2)C cells, ACBI1 treatment (24 h) resulted in decreased chromatin accessibility (>2.5-fold changes) in 37,425 peaks (28.04%) (Fig. 2A, middle panel), while 28.12% remained unaffected by ACBI1 treatment (within 1.25-fold changes). Furthermore, genetic silencing of *SMARCA4* using two different siRNAs in IMR5 cells also led to significant decreases (>2-fold) in chromatin accessibility in 16,318 peaks (13.11%) for both siRNAs (Fig. 2A, right panel), while 43.71% remained unaffected by genetic

silencing of *SMARCA4* (within 1.25-fold changes). HOMER analysis of motifs associated with reduced chromatin accessibility after SWI/SNF ATPases depletion revealed enrichment of non-canonical E-boxes (CAXXGT) known to be bound by basic helix-loop-helix TFs including MYCN and HAND2 (Xu et al, 2023), as well as DNA binding motifs of NB CRC TFs PHOX2B, GATA3 and MEIS2 (Boeva et al, 2017; De Wyn et al, 2021) (Fig. 2B; Appendix Fig. S2A). Gene ontologies (GO) analysis by using the genomic regions enrichment of annotation tool (GREAT) (Mootha et al, 2003; Subramanian et al, 2005) showed that the genes associated with reduced chromatin accessibility were enriched in pathways related to axon guidance and semaphorin-plexin signaling (Fig. 2C; Appendix Fig. S2B). Further analysis of peak distribution by using the GREAT and HOMER tools demonstrated that >95% of the compacted sites primarily localized to intergenic and intronic regions, more than 5 kb away from the transcriptional start sites (TSS) (Fig. 2D; Appendix Fig. S2C). These findings indicate that SWI/SNF ATPases depletion resulted in the compaction of distal gene regulatory regions rather than promoter regions.

## Depletion of SWI/SNF ATPases dislodges core TFs from DNA at the compacted chromatin sites

Concurrent with the loss of chromatin accessibility on cis-regulatory elements bound by core TFs, ChIP-seq revealed decreases in the average signals of HAND2, PHOX2B, GATA3, MYCN, and H3K27ac at the compacted sites after pharmacological targeting of SWI/SNF ATPases with ACBI1 in IMR32 cells, shown by metagene plots (Fig. 2E). Genetic silencing of *SMARCA4* also resulted in decreased average ChIP-seq signals of these core TFs and H3K27ac at the compacted sites (Fig. 2F), although to a lesser extent. This may be due to the lower efficacy of genetic silencing compared to ACBI1 treatment at depleting SMARCA4 protein (Appendix Fig. S2D,E). It is also possible that the upregulation of SMARCA2 after *SMARCA4* knockdown in IMR32 cells (Appendix Fig. S1I, middle panel) plays a compensatory role that affects the ChIP-seq signals of H3K27ac and core TFs.

To investigate whether the reduction in DNA binding primarily occurred for core oncogenic TFs following SWI/SNF ATPase depletion in NB cells, we conducted additional ChIP-seq experiments for the cohesin subunit RAD21, the transcription repressor CTCF, and the histone-methylating complex subunit WDR5. Metagene plots (Appendix Fig. S2F) revealed decreased average signals for all these proteins at the compacted sites after targeting SWI/SNF ATPases with ACBI1 in IMR32 cells.

To determine whether the reduction in DNA binding of the core TFs, H3K27ac, RAD21, CTCF, and WDR5 occurs exclusively at sites with decreased chromatin accessibility, we assessed their binding at locations where ATAC-seq signals remained unchanged following ACBI1 treatment. Our findings revealed decreased average ChIP-seq signals for HAND2, PHOX2B, and H3K27ac, increased signals for GATA3 and MYCN, while there was little to no change in the signals for RAD21, CTCF, and WDR5 (Appendix Fig. S2G). The western blot assay indicated that ACBI1 treatment reduced PHOX2B and HAND2 protein levels by ~30%, and reduced H3K27ac levels by 15% (Appendix Fig. S2D), which possibly contributes to the decrease in their ChIP-seq signals at both sites with stable and decreased chromatin accessibility. Interestingly, despite no change in protein levels detected by

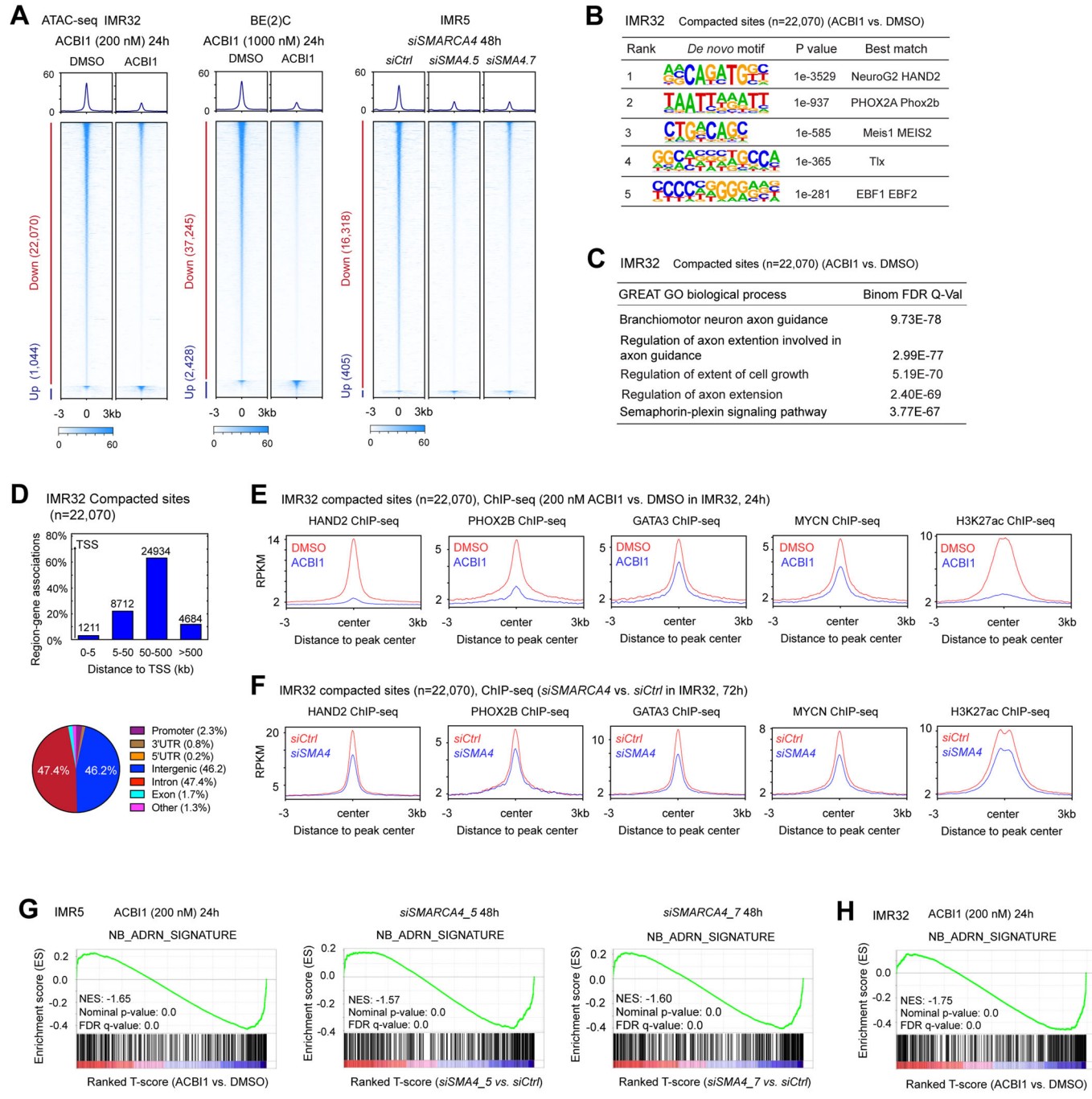

**Figure 2. Depletion of SWI/SNF ATPases decreases chromatin accessibility and disrupts core TFs from binding to DNA.**

(**A**) Analysis of differential chromatin accessibility reveals that depletion of both SMARCA2 and SMARCA4, or SMARCA4 alone, leads to reduced chromatin accessibility, as indicated by decreased ATAC-seq signals in the heatmaps. (**B**) HOMER de novo motif scan of sites with reduced chromatin accessibility after ACBI treatment in IMR32 cells. (**C**) GREAT GO analysis of ATAC-seq peaks with reduced signal intensities. (**D**) ATAC-seq peak distribution analysis reveals that sites with reduced chromatin accessibility are enriched in the distal regulatory regions. (**E, F**) Metagene plots illustrate the decreased average ChIP-seq signals in HAND2, PHOX2B, GATA3, MYCN, and H3K27ac at the sites with reduced chromatin accessibility after ACBI1 treatment or genetic silencing of SMARCA4. (**G**) Gene set enrichment analysis (GSEA) of RNAs-seq data shows negative enrichment of genes highly expressed in ADRN-type NB following treatment with ACBI1 for 24 h, or genetic silencing of SMARCA4 for 48 h using two different siRNAs in IMR5 cells. (**H**) GSEA shows negative enrichment of genes highly expressed in ADRN-type NB following treatment with ACBI1 for 24 h in IMR32 cells. Data information: In panel (**B**), HOMER employs the Hypergeometric Test to determine the statistical significance of the overlap between a set of observed sequences (e.g., DNA sequences containing motifs) and a set of background sequences. In panels (**G**) and (**H**), GSEA uses permutation testing to estimate the significance of the enrichment score.

western blot (Appendix Fig. S2D), ACBI1 treatment increased MYCN and GATA3 ChIP-seq signals at the sites without changes in ATAC-seq signals. We postulate that this effect may be due to the re-localization of MYCN and GATA3 from compacted sites to open chromatin.

In NB, core TFs including MYCN, HAND2, PHOX2B, and GATA3 cooperatively regulate the transcriptional profile that defines the ADRN-type of NB (Boeva et al, 2017; Durbin et al, 2018; van Groningen et al, 2017; Xu et al, 2023). The finding that targeting SWI/SNF ATPases leads to the compaction of *cis*-regulatory elements bound by core TFs and dislodges these core TFs from DNA, would predict the loss of the ADRN cell transcriptional signature upon depletion of SWI/SNF ATPases. As expected, gene set enrichment analysis (GSEA) of the RNA-seq data (Dataset EV1) showed that in IMR5 cells, ACBI1 treatment for 8 h or 24 h resulted in a negative enrichment of ADRN signature genes derived from ADRN-type NB(van Groningen et al, 2017) (Fig. 2G; Appendix Fig. S2H). Genetic silencing of *SMARCA4* in IMR5 cells using two different *siRNAs* (Dataset EV1) also resulted in a negative enrichment of ADRN signature genes (Fig. 2G). Similarly, analysis of RNA-seq data (Dataset EV2) from ACBI1-treated IMR32 cells showed a negative enrichment of ADRN signature genes (Fig. 2H). Notably, the GSEA results of SWI/SNF ATPases depletion exhibited a bimodal pattern. This is because, although more ADRN signature genes are down-regulated, some of the ADRN signature genes are up-regulated (Dataset EV3). Despite this bimodal pattern, the GSEA statistical analysis of the overall pattern indicated significant negative enrichment of ADRN signature genes (Fig. 2G,H; Appendix Fig. S2H).

## SWI/SNF ATPases drive invasive transcriptional program and depletion of SWI/SNF ATPases suppresses NB invasion

GSEA GO analysis revealed that the top-ranked negatively enriched gene sets, after the depletion of SWI/SNF ATPases in either IMR5 or IMR32 cells, were related to collagen-containing extracellular matrix and external encapsulating structure (Fig. 3A; Appendix Fig. S3A–D). These genes are known to play roles in cell growth, movement, and invasion, suggesting that the depletion of SWI/SNF ATPases was linked to changes in the invasiveness of NB cells. Moreover, GSEA GO analysis indicated the top-ranked positively enriched gene sets after the depletion of SWI/SNF ATPases in IMR5 cells were ribosomal protein-encoding genes (Dataset EV4). After the depletion of SWI/SNF ATPases in IMR32 cells, the top-ranked positively enriched gene sets were related to DNA repair complex and lens fiber cell differentiation. However, none of these pathways showed significant enrichment (considered significant at $p < 0.05$, FDR q < 0.25) (Dataset EV4). These results indicate that the depletion of SWI/SNF ATPases leads to similar negatively enriched gene sets, but it does not result in similar positively enriched gene sets across different NB cell lines.

To investigate the biological function of SWI/SNF ATPases in regulating cell migration and invasion in NB, we performed the IncuCyte cell migration scratch wound assay and IncuCyte single spheroid matrigel invasion assays. Treatment of IMR5 or BE(2)C cells with ACBI1 and AU-15330 resulted in significant inhibition of cell movement in the scratch wound assay (Fig. 3B,C; Appendix Fig. S3E). Furthermore, treatment of NB cells with ACBI1 or AU-15330 led to

almost complete inhibition of single spheroid matrigel cell invasion in IMR5-luc-GFP, IMR32-luc-GFP, and in BE(2)C-luc-GFP (Fig. 3D–F). Notably, this treatment also halted the spheroid expansion of IMR5 and IMR32 cells within the matrigel and reduced the spheroid expansion of BE(2)C cells (Fig. 3D–F). As ACBI1 or AU-15330 induction of cell death may contribute to the decreases in cell invasion, we assessed cell death in these spheroid cultures using an LDH-Glo Cytotoxicity Assay, as well as monitoring the cellular expression of GFP as has been shown in other studies (Steff et al, 2001). ACBI1 and AU-15330 did not increase cell death in BE(2)C, but cell death increased in IMR5 and IMR32 cells in the LDH-Glo Cytotoxicity Assay (Fig. S3F). All NB spheroids retained GFP expression even after 14 days of ACBI1 or AU-15330 treatment, comparable to DMSO-treated spheroids (Appendix Fig. S3G). This suggests that despite the observed increase in cell death in IMR5 and IMR32 cells detected by the LDH-Glo cytotoxicity assay, many cells remain viable. Overall, the reduction of matrigel invasion following ACBI1 or AU-15330 treatment may be partially attributed to cell death in IMR5 and IMR32 cells, but not in BE(2)C cells. Our results demonstrate that the depletion of SWI/SNF ATPases in NB cells not only suppresses proliferation but also decreases their migratory and invasive capabilities. Importantly, this finding is clinically relevant, as levels of *SMARCA4* mRNA were significantly elevated in the metastatic NB cases, namely the high-risk, recurrent, or progressed patient NB, compared to the other patient NB cases (Appendix Fig. S3H). Additionally, *SMARCA4* expression was found to be significantly higher in *MYCN*-amplified than non-amplified tumors, which are commonly seen in patients with disseminated disease (Appendix Fig. S3I). These findings suggest that SWI/SNF ATPases contribute to the invasive behavior of NB.

## SWI/SNF ATPases establish permissive chromatin environments for MES genes in ADRN-type NB

In addition to the suppression of invasive cell signatures (Fig. 3A; Appendix Fig. S3A–D), depletion of SWI/SNF ATPases revealed a negative enrichment of genes associated with the hallmark epithelial-mesenchymal transition (EMT) in IMR5 cells (Appendix Fig. S4A). This observation suggested that SWI/SNF ATPases might be involved in the regulation of MES signature genes derived from MES-type NB (van Groningen et al, 2017). To our surprise, despite being classified as an ADRN-type NB, the GSEA results indicated a significant negative enrichment of this MES gene signature after SWI/SNF ATPases depletion in IMR5 (Appendix Fig. S4B). Similar results were observed also in IMR32 cells (Appendix Fig. S4C). As shown in the heatmap (Fig. 4A), depleting SWI/SNF ATPases through ACBI1 treatment for 8 or 24 h, or via genetic silencing using two different *SMARCA4 siRNAs*, consistently led to the down-regulation of representative ADRN genes, MES genes, EMT genes, and extracellular matrix protein-encoding genes. This finding strongly supports a model in which SWI/SNF ATPases not only drive ADRN gene expression but also influence the expression of MES genes in ADRN-types of NB cells, thus potentially contributing to the intrinsic plasticity of NB cells.

Interconversion between the undifferentiated MES subtype and the predominantly committed ADRN subtype of NB is proposed to contribute to intra-tumoral heterogeneity and the plasticity of NB cells (Thirant et al, 2023; van Groningen et al, 2017). This is illustrated in the Waddington landscape model (Fig. 4B). By

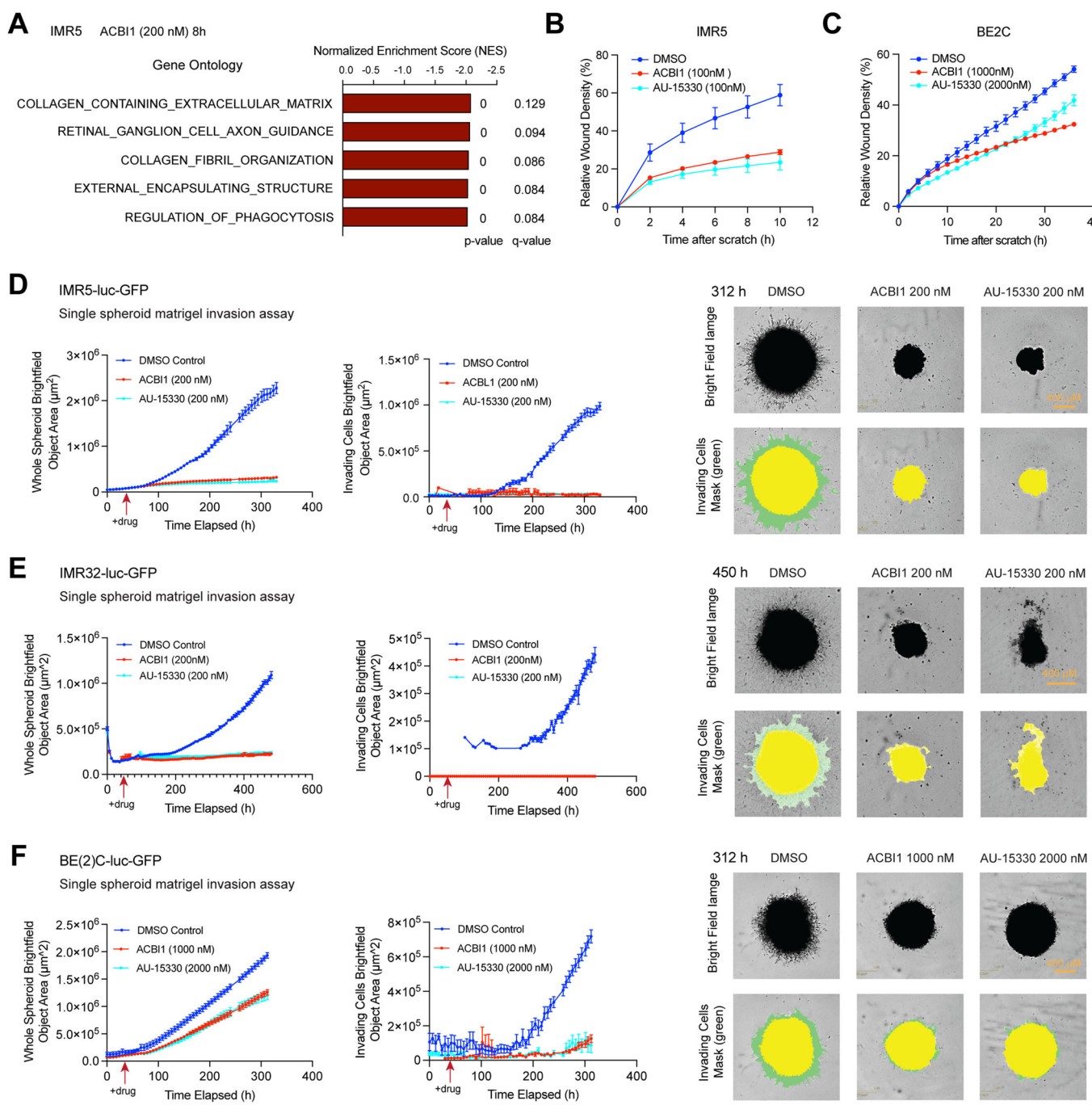

**Figure 3. SWI/SNF ATPases drive invasive transcriptional program and depletion of SWI/SNF ATPases suppresses NB invasion.**

(A) GSEA gene ontologies (GO) analysis reveals negative enrichment of gene sets related to collagen-containing extracellular matrix and collagen fibril organization after 8 h ACBI1 treatment of IMR5 cells. (B, C) IncuCyte scratch wound healing assay demonstrates a significant decrease in relative would density upon treatment of IMR5 and BE(2)C with ACBI1 or AU-15330 ($n = 4$/group; Error bars indicate SEM). Data are representative of two independent experiments. (D–F) IncuCyte single spheroid matrigel invasion assay shows a significant decrease of spheroid size and invasion upon treatment of IMR5-luc-GFP, IMR32-luc-GFP and BE(2)C-luc-GFP cells with ACBI1 or AU-15330, as indicated by both bar graphs and images ($n = 4$ or 6/group; Error bars indicate SEM). Data are representative of three independent experiments. Data information: In panel (A), GSEA uses permutation testing to estimate the significance of the enrichment score. Source data are available online for this figure.

comparing the RNA read counts of ADRN genes and MES genes in the ADRN-type NB cell line IMR32, we found that 94% of ADRN genes (345/369 genes) and 82% of MES genes (397/485) were detectable at the mRNA level, although the basal mRNA levels of MES genes were significantly lower than those of ADRN genes

(Fig. 4C). This finding implies that the simultaneous expression of ADRN and MES signature genes in adrenergic NB cells plays a role in the inherent plasticity of ADRN-type NB cells.

Permissive chromatin establishes a state of "epigenetic plasticity" and chromatin accessibility at regulatory chromatin precedes

   

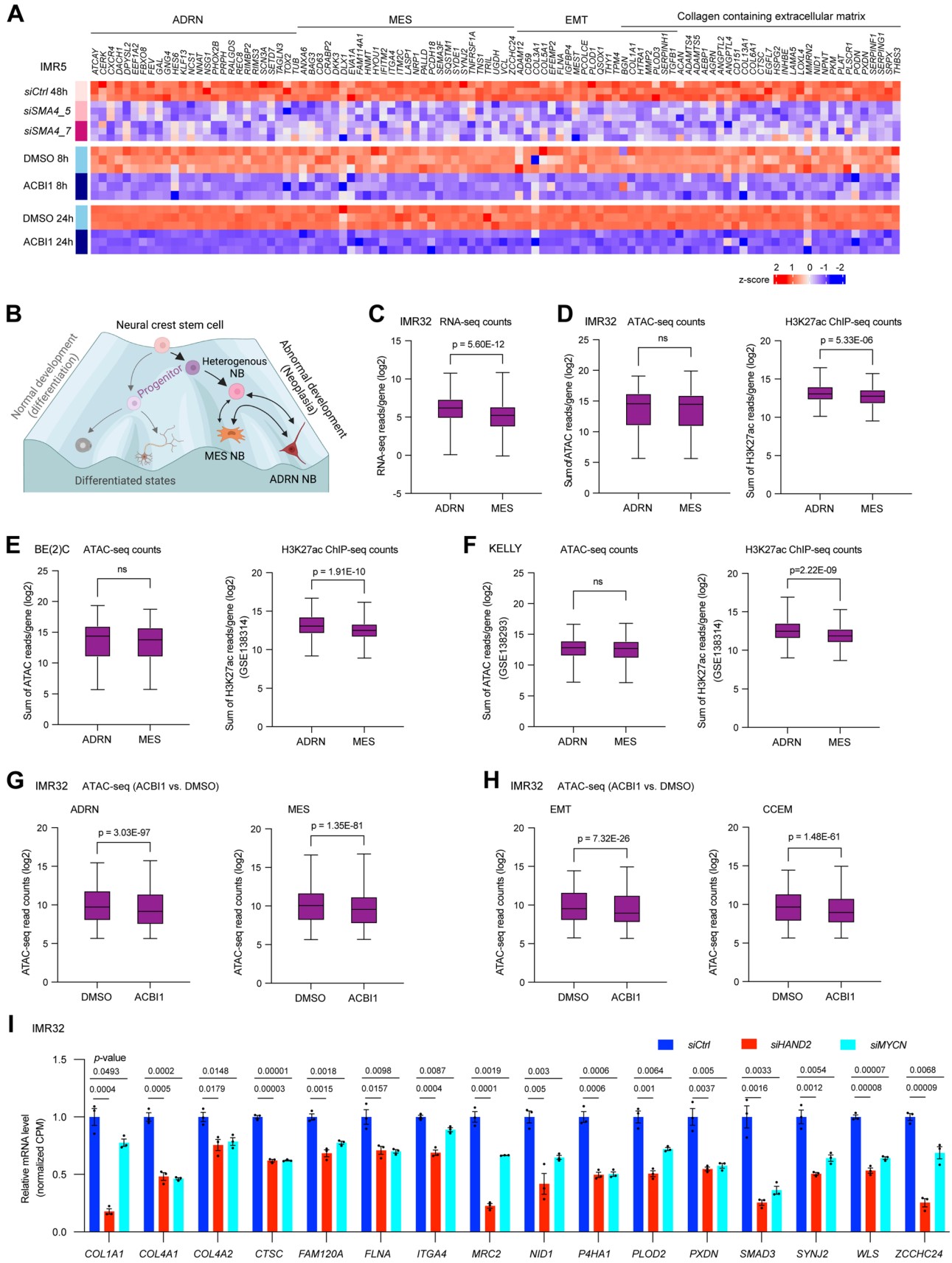

◄

**Figure 4. SWI/SNF ATPases maintain permissive chromatin state for MES genes in ADRN-type NB.**

(A) Heatmaps display representative genes that are significantly down-regulated ($p < 0.05$) after genetic silencing of *SMARCA4* using two different siRNAs or 8 h or 24 h treatment of IMR5 cells with ABCl1. (B) Waddington landscape depicting the plasticity of NB, where the heterogenous NB shown in the cartoon could represent a transitional NB cell population. (Created with BioRender.com). (C) RNA-seq read counts reveal significantly lower basal mRNA levels of MES genes ($n = 397$; Error bars indicate SEM) compared to ADRN genes ($n = 345$; Error bars indicate SEM), but MES gene expression is detectable in the ADRN-type NB cell line IMR32. (D) ATAC-seq read counts demonstrate similar chromatin accessibility of MES genes ($n = 6112$ peaks of 373 genes; Error bars indicate SEM) compared to ADRN genes ($n = 5385$ peaks of 276 genes; Error bars indicate SEM), while H3K27ac ChIP-seq signals on MES genes ($n = 726$ peaks of 276 genes; Error bars indicate SEM) are significantly lower than on ADRN genes ($n = 1103$ peaks of 291 genes; Error bars indicate SEM) in IMR32 cells. (E) ATAC-seq read counts demonstrate similar chromatin accessibility of MES genes ($n = 6065$ peaks of 382 genes; Error bars indicate SEM) compared to ADRN genes ($n = 6672$ peaks of 287 genes; Error bars indicate SEM), while H3K27ac ChIP-seq signals on MES genes ($n = 1468$ peaks of 348 genes; Error bars indicate SEM) are significantly lower than on ADRN genes ($n = 2571$ peaks of 331 genes; Error bars indicate SEM) in BE(2)C cells. (F) ATAC-seq read counts demonstrate similar chromatin accessibility of MES genes ($n = 7957$ peaks of 460 genes; Error bars indicate SE) compared to ADRN genes ($n = 6211$ peaks of 349 genes; Error bars indicate SEM), while H3K27ac ChIP-seq signals on MES genes ($n = 1453$ peaks of 355 genes; Error bars indicate SEM) are significantly lower than on ADRN genes ($n = 2511$ peaks of 338 genes; Error bars indicate SEM) in KELLY cells. (G, H) Analysis of ATAC-seq data reveals a significant decrease in chromatin accessibility of ADRN ($n = 1779$; Error bars indicate SEM), MES ($n = 1508$; Error bars indicate SEM), EMT ($n = 601$; Error bars indicate SEM), and collagen-containing extracellular matrix (CCEM) genes ($n = 961$; Error bars indicate SEM) that are down-regulated after ACBI1 treatment of IMR32 cells. (I) Representative MES genes that are down-regulated after SWI/SNF ATPases depletion are also found to be down-regulated after the silencing of *HAND2* or *MYCN* in IMR32 cells from three biological replicates ($n = 3$; Error bars indicate SEM). Data information: In panels (C–H), data are presented as box plots, where the middle solid lines indicate the mean, and the whiskers represent min to max. In panels (C–F) and (I) the data are represented as mean ± SEM. Statistical differences were calculated using a two-sided unpaired Student's *t*-test, where "ns" represents not significant. In panels (G, H), the data represent mean ± SEM, and statistical differences were calculated using a two-sided paired Student's *t*-test.

gene expression to ensure a timely and cell-specific gene expression in response to external stimuli, which may occur during lineage specification and cancer initiation (Bonifer and Cockerill, 2017; Flavahan et al, 2017; Ma et al, 2020; Vicente-Duenas et al, 2018). ADRN-type NB cells can give rise to heterogeneous cell populations comprising both ADRN- and MES-type cells in the in vitro cell culture environment (Thirant et al, 2023; van Groningen et al, 2017). To investigate whether the high plasticity of ADRN-type NB cells is due to permissive chromatin, we analyzed the chromatin accessibility of ADRN genes and MES genes using ATAC-seq data in IMR32. We found no statistical difference in chromatin accessibility, based on the sum of the ATAC-seq read counts per gene between ADRN genes and MES genes (Fig. 4D, left panel). H3K27ac ChIP-seq signals on MES genes were significantly lower than those on ADRN genes (Fig. 4D, right panel), which is consistent with previous research indicating that some of the ADRN genes but not MES genes are driven by super-enhancers in ADRN-type NB cells (Boeva et al, 2017; van Groningen et al, 2017). Similar results were observed in two other ADRN-type NB cell lines BE(2)C and KELLY after analysis of our data and publicly available ATAC-seq (GSE138293) and H3K27ac ChIP-seq (GSE138314) data (Upton et al, 2020) (Fig. 4E,F). Depletion of SWI/SNF ATPases led to a significant decrease in ATAC-seq signals on both ADRN genes and MES genes (Fig. 4G), as well as on EMT genes and collagen-containing extracellular matrix (CCEM) genes (Fig. 4H). These results indicate that SWI/SNF ATPases enable comparable chromatin accessibility in both ADRN genes and MES genes even in predominantly ADRN-type NB cells. Moreover, SWI/SNF ATPases play a regulatory role in the chromatin accessibility of these genes.

To identify potential TFs binding to MES genes and regulating their expression in ADRN-type NB cells, we performed HOMER motif scan analysis of ATAC-seq peaks associated with MES genes. Surprisingly, we found that adrenergic core TFs such as PHOX2B and ISL1 binding motifs, along with non-canonical E-boxes were among the top enriched motifs at MES genes in the ADRN-type IMR32 or KELLY cell lines (Appendix Fig. S4D,E). Notably, the core adrenergic TFs HAND2, MYCN, and ASCL1 are known to bind to non-canonical E-boxes. Additionally, we conducted

GREAT GO analysis of the ATAC-seq peaks and observed that peaks associated with ADRN genes were enriched in neuron development, while peaks associated with MES genes were enriched in extracellular matrix organization in both IMR32 and KELLY cells (Appendix Fig. S4F,G). Supporting the enrichment of adrenergic core TF binding motifs in MES genes, we identified ChIP-seq peaks of HAND2, MYCN, PHOX2B, and GATA3 on MES genes, albeit with lower ChIP-seq signal intensities compared to ADRN genes (Appendix Fig. S4H). Moreover, RNA-seq data (GSE183641) (Xu et al, 2023) analysis revealed that silencing of *HAND2* and *MYCN* led to decreased expression of a subset of MES genes, which were also down-regulated after SWI/SNF ATPases depletion (Fig. 4I). These observations suggest that SWI/SNF ATPases play a role in enabling adrenergic core TFs to bind and maintain basal MES gene expression, potentially adding to the inherent plasticity of ADRN-type NB cells.

## Depletion of SWI/SNF ATPases reduces NB cell heterogeneity and plasticity

Previous studies have shown that NB cells derived from the same patient or patient-derived xenograft (PDX) exhibit intra-tumoral heterogeneity, with MES-type cells forming attached monolayers and exhibiting lamellipodia, while ADRN-type cells form semi-attached spheres (Persson et al, 2017; Thirant et al, 2023; van Groningen et al, 2017). To investigate the impact of depleting SWI/SNF ATPases on the emergence of heterogenous NB subpopulations, we used the SJNBL012407_X1 PDX cell model (*MYCN*-amplified). Consistent with previous findings, sphere formation occurred when SJNBL012407_X1 cells were cultured in neural stem-cell culture media (SCM) (Fig. 5A; Appendix Fig. S5A). Upon switching the cells to complete media containing 10% fetal bovine serum (FBS), the growth of MES-morphology cells in a monolayer and ADRN-type cells in semi-attached spheres became evident (Fig. 5A; Appendix Fig. S5A). When cultured in FBS-containing media, treatment with ACBI1 resulted in the depletion of SWI/SNF ATPases (Appendix Fig. S5B). In the DMSO-treated group (vehicle control), there was a substantial increase in the number of flattened, enlarged, MES-type morphology monolayer cells over time after

switching the media from SCM to FBS-containing media (Appendix Fig. S5C, Movie EV1). However, the ACBI1-treated group exhibited a reduced cell count, as evidenced by both captured cell images and quantified Incucyte cell confluence assay (Fig. 5A,B). ACBI1 treatment impedes the expansion of flattened, enlarged, monolayer cells over time, which can be more clearly visible in the uncropped cell images (Appendix Fig. S5C), and the associated Incucyte movies (Movies EV1 and EV2). Moreover, we found that ACBI1 treatment of SJNBL012407_X1 cells cultured in SCM also significantly suppressed cell proliferation (Appendix Fig. S5D), indicating that SWI/SNF ATPases are essential for PDX cells cultured in either SCM or FBS-containing media.

To investigate whether cellular clones are selected due to toxicity, we conducted a cell cytotoxicity assay using the Incucyte Cytotox Green Dye to evaluate the impact of ACBI1 treatment on SJNBL012407 cell death. Our observations revealed increased cell death, but there was less than 10% cell death in the ACBI1-treated cultures (Appendix Fig. S5E,F). The cell death was observed in both ADRN-like large spheres and cells outside of the large spheres (Appendix Fig. S5E). Furthermore, after SJNBL012407_X1 cells formed heterogeneous populations following 3 days of culture in FBS-containing media, the administration of ACBI1 could not completely deplete MES-type cells (Appendix Fig. S5G), although it significantly decreased cell proliferation (Appendix Fig. S5H). These findings indicate that the depletion of SWI/SNF ATPases impairs the ability of PDX cells to adopt a heterogenous morphology when transitioning from SCM to FBS-containing media. Suppression of cell proliferation and induction of cell death may contribute to this impairment.

To investigate the impact of depleting SWI/SNF ATPases at a molecular level on SJNBL012407_X1 cell plasticity, we performed transcriptomic analysis. Initially, we compared the transcriptional profiles of SJNBL012407_X1 cells cultured in FBS-containing media for 5 days to those cultured in SCM. The bulk RNA-seq data (Dataset EV5) analysis revealed a negative enrichment of genes highly expressed in Schwann cell precursors (SCP) and neural stem cells (NSC), while ADRN and MES signature genes exhibited positive enrichment (Appendix Fig. S5I,J). Next, we evaluated the impact of SWI/SNF ATPases depletion on gene expression in SJNBL012407_X1 cells cultured in FBS-containing media. Similar to findings in other NB cell lines, ACBI1 treatment of SJNBL012407_X1 cells resulted in a negative enrichment of ADRN, MES, EMT, and collagen-containing extracellular matrix (CCEM) signature genes (Appendix Fig. S5K,L). Moreover, 4 days of ACBI1 treatment led to a positive enrichment of genes associated with immature dopaminergic neurons (HDA) and genes associated with outer radial glial cells undergoing neuronal differentiation (Appendix Fig. S5M).

We conducted single-cell RNA-seq (scRNA-seq) analysis to investigate how loss of SWI/SNF ATPases affected gene expression as SJNBL012407_X1 cells were shifted from culture in SCM to FBS-containing media in the absence or presence of ACBI1 (4 days treatment). Integrative analysis of all 10488 cells from the three groups identified fourteen distinct cell clusters highlighted by different feature genes (Dataset EV6). While all these clusters could be observed in FBS-cultured cells, only six of them (clusters 0, 1, 2, 3, 4, and 6) could be detected in SCM-cultured cells (Fig. 5C, Dataset EV7), indicating the possibility of the other seven clusters consisting of cells that are uniquely induced by the FBS culturing

condition, whereas the ACBI1 treatment did not affect the unsupervised clustering of cells presented in FBS-containing media (Dataset EV7). Thus, we specifically focused on those altered signatures in response to ACBI1 treatment identified by bulk-RNA seq. By analyzing ADRN and MES cell signatures in the scRNA-seq data, we found a notable increase in the ADRN signature score upon switching the cell culture media from SCM to FBS-containing media, while there was a lower ADRN signature score in ACBI1-treated cells. This trend was evident from the changes in the intensity of cells with high ADRN signature scores shown in the UMAP plot and was supported by the statistical analysis of the average ADRN signature score per cell (Fig. 5D). Moreover, the average MES signature score per cell was significantly higher in cells cultured in FBS-containing media (DMSO group) compared to cells cultured in SCM, and ACBI1 treatment resulted in a significant reduction in the MES signature score (Fig. 5E). Importantly, our results demonstrate that the reduction of MES gene expression after ACBI1 treatment prevents the development of MES morphology cells from PDX cells when cultured in FBS-containing media, as shown in Fig. 5A.

Bulk-RNA seq revealed a positive enrichment of genes associated with neuronal differentiation following ACBI1 treatment of SJNBL012407_X1 cells cultured in FBS-containing media (Appendix Fig. S5M). To investigate the impact of SWI/SNF ATPases depletion on the neuronal differentiation of SJNBL012407_X1 cells at the single-cell level, we further analyzed the scRNA-seq data. Specifically, we focused on the up-regulated neuronal differentiation signatures identified in ACBI1-treated cells based on bulk RNA-seq (Dataset EV8). In the UMAP plots (Appendix Fig. S5N,O), we observed that more SJNBL012407_X1 cells exhibited high HDA (dopaminergic signature) and neuronal differentiation signature scores in ACBI1-treated cells cultured in FBS-containing media, compared to DMSO-treated cells cultured in FBS-containing media or cells cultured in SCM. These results indicate that while ACBI1 treatment of SJNBL012407_X1 cells cultured in FBS-containing media resulted in ADRN and MES scores comparable to the cells cultured in SCM (Fig. 5D,E), the depletion of SWI/SNF ATPases dramatically increased the population of cells undergoing neuronal differentiation and suppressed cell proliferation.

## Depletion of SWI/SNF ATPases sensitizes NB cells to chemotherapeutic drug treatment

MES-type NB cells are enriched after chemotherapy and in relapsed patients, and they are known to be more resistant to many chemotherapeutic drug treatments (Boeva et al, 2017; Gartlgruber et al, 2021; Gautier et al, 2021; Ponzoni et al, 2022; van Groningen et al, 2017). We found that depletion of SWI/SNF ATPases reduces the proportion of MES-type cells in SJNBL012407_X1, which suggests that the targeting of SWI/SNF ATPases may inhibit the development of chemoresistance. To test this, we treated SJNBL012407_X1 cells with or without ACBI1. After 4 days, the cells were treated with or without etoposide, a cytotoxic drug used in the treatment of NB patients. Western blotting indicated a 50% reduction in SMARCA4 protein levels after 40 nM ACBI1 treatment and a greater than 90% reduction with 250 nM treatment (Appendix Fig. S6A). The heatmap displayed a variable dose response of the cells to ACBI1 and etoposide treatment (Fig. 6A).

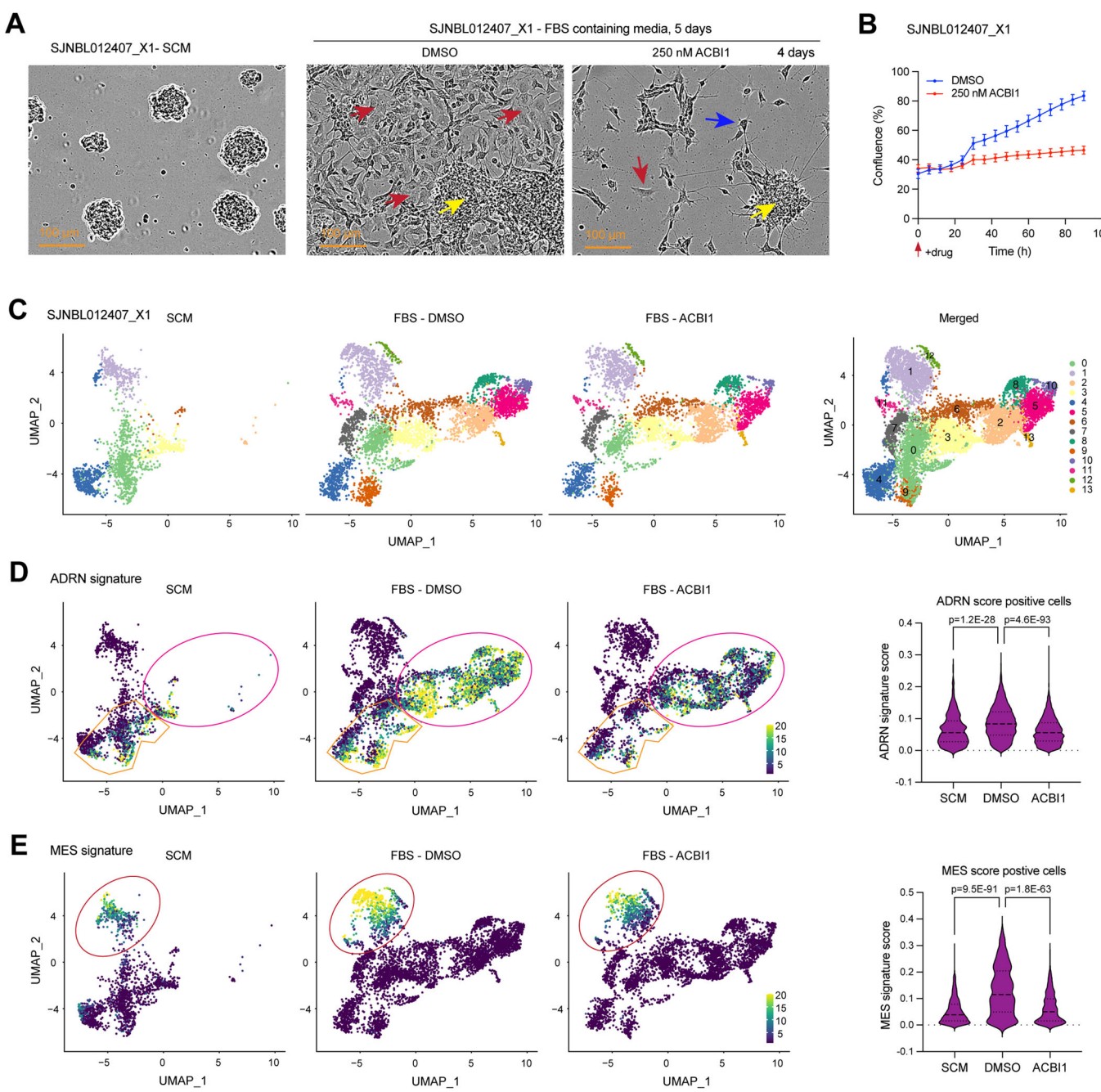

**Figure 5. SWI/SNF ATPases depletion reduces NB cell heterogeneity and plasticity.**

(A) Representative images of SJNBL012407_X1 PDX cells cultured in neural stem-cell culture media (SCM) or shifted to culture in complete media with 10% fetal bovine serum (FBS), and after 16 h the cells were treated with DMSO (vehicle control) or ACBI1 for 4 days. Cell images reveal a distinct enlargement and flattening of mesenchymal-like morphology monolayer cells in FBS-containing media, DMSO-treated SJNBL012407_X1 cells (red arrows indicate the expanded area and adjacent cells). In contrast, ACBI1-treated cells (red arrow indicates MES-morphology type cells) predominantly exhibit spheres and neuroblast-like cells, with the rare presence of flattened, enlarged MES-morphology monolayer cells. The yellow arrow points to the semi-attached sphere. The blue arrow points to the representative neuroblast-like cell. (B) Incucyte cell confluence assays show that ACBI1 treatment of SJNBL012407_X1 reduces cell proliferation (% confluence) ($n = 3$; Error bars indicate SEM). Data are representative of three independent experiments. (C) Single-cell RNA-seq analysis revealed six clusters for SJNBL012407_X1 cells cultured in SCM and thirteen clusters for cells cultured in FBS-containing media. (D, E) UMAP plots show ADRN and MES signature scores in SJNBL012407_X1 cells under indicated culture conditions. Left panels: Gene signature score high cells are indicated by green and yellow dots, and circles highlight dominant cell clusters with high gene signature scores observed in at least one cell culture condition; Right panel: statistical analysis of the average ADRN (cell number: SCM, $n = 790$; DMSO, $n = 3307$; ACBI1, $n = 2467$) or MES (cell number: SCM, $n = 880$; DMSO, $n = 813$; ACBI1, $n = 876$) signature score per cell under different conditions. Data information: In the right panels of (D, E), data are presented as violin plots, where the dashed lines indicate the median and the 25th and 75th percentiles. Statistical differences were calculated using a two-sided unpaired Student's $t$-test. Source data are available online for this figure.

 

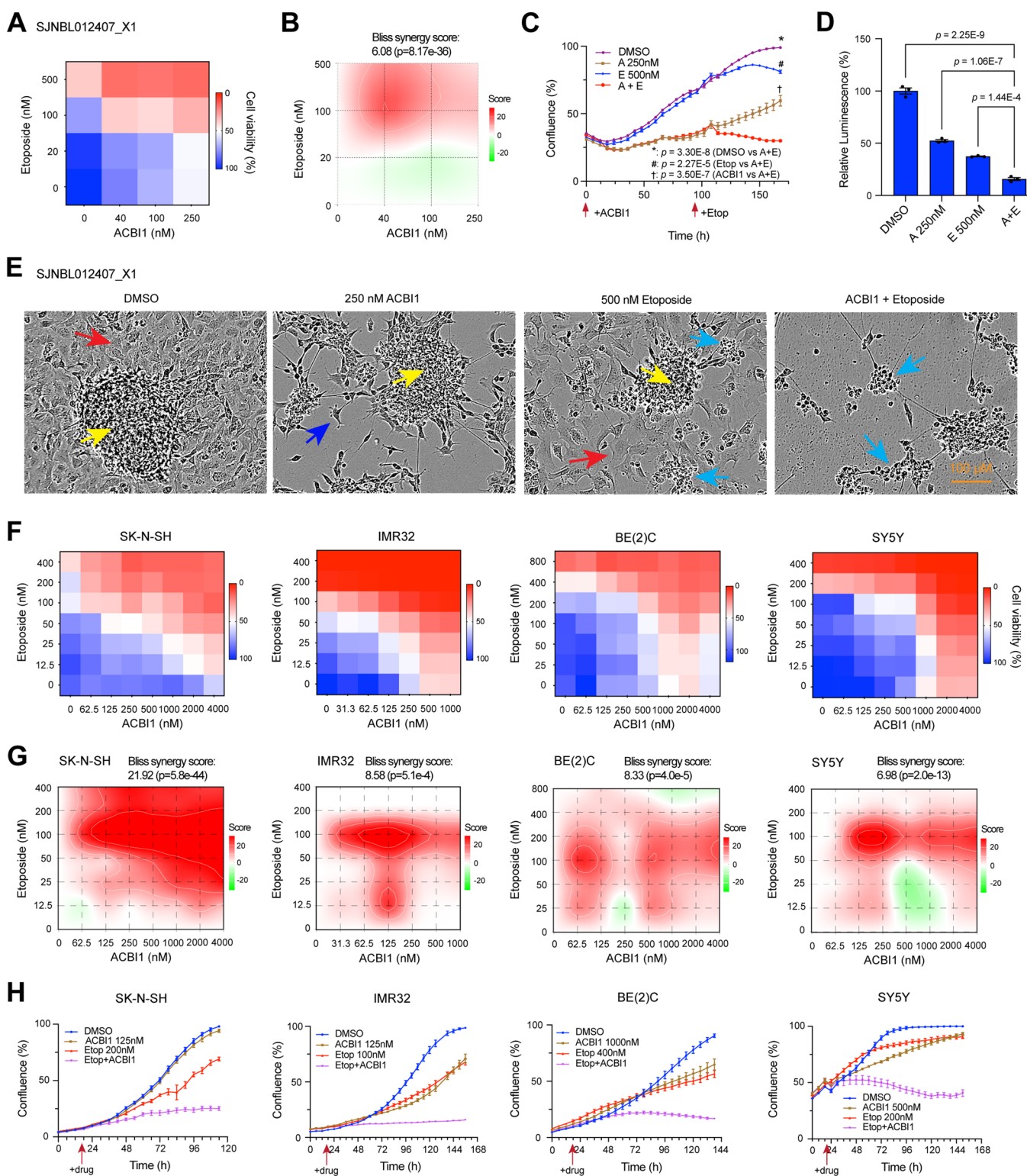

Importantly, combining ACBI1 and etoposide led to a synergistic reduction in viable cell numbers, with average bliss synergy scores greater than 10 across a range of doses (Fig. 6B). Cell confluence assays (Fig. 6C) and cell viability assays (CellTiter-Glo) (Fig. 6D) revealed that single treatments of 250 nM ACBI1 or 500 nM etoposide alone was less effective than the combination treatment.

As anticipated, the enlarged representative cell images (Fig. 6E) and uncropped cell images (Appendix Fig. S6B) demonstrated that ACBI1-treated cells predominantly exhibit spheres and neuroblast-like cells, with fewer flattened monolayer cells. When SJNBL012407_X1 cell line is exposed to etoposide alone, there was a reduction in ADRN-type cell numbers, but a substantial

**Figure 6. Sensitization of NB cells to chemotherapeutic drug treatment through depletion of SWI/SNF ATPases.**

(A) Heatmap displays the percentage of cell viability following varying doses of ACBI1 and etoposide treatment in SJNBL012407_X1 PDX cells ($n = 3$). The cells are cultured in FBS-containing media and initially subjected to either ACBI1 treatment alone or no treatment for 4 days. Subsequently, they are exposed to etoposide with or without prior ACBI1 treatment for an additional 3 days. Cell viabilities are measured by CellTiter-Glo Cell Viability Assay. (B) SynergyFinder online tool is used for bliss synergistic analysis to evaluate the synergistic effect of ACBI1 and etoposide treatment in SJNBL012407_X1 cells ($n = 3$). (C) Incucyte cell confluence assays to measure the effect of 250 nM ACBI1 or 500 nM etoposide, or ACBI1 plus etoposide treatment on cell proliferation (% confluence) ($n = 3$ wells/group; Error bars indicate SEM). (D) CellTiter-Glo assay measures the effect of 250 nM ACBI1 or 500 nM etoposide, or ACBI1 plus etoposide treatment on cell growth ($n = 3$ wells/group; Error bars indicate SEM). Cell viabilities of DMSO-treated cells are set to 100%, and the bar graph is generated using GraphPad Prism software. (E) Representative images of SJNBL012407_X1 PDX cells treated with DMSO (a vehicle control), 250 nM ACBI1, 500 nM etoposide, and ACBI1 + etoposide are presented. These images are captured using the IncuCyte SX5 imaging system. Cell images reveal distinct changes, with DMSO-treated cells showing spheres, as well as enlargement and flattening of MES-type monolayer cells. ACBI1-treated cells predominantly exhibit spheres and neuroblast-like cells, with a rare presence of flattened, enlarged MES-type cells. Etoposide-treated cells exhibit a reduced number of ADRN-type cells, but a substantial population of MES-type cells persists in the cultures. In contrast, cells treated with a combination of ACBI1 and etoposide exhibit predominantly unhealthy or decreased cells, indicated by the altered cell morphology. The yellow arrow points to the sphere, the red arrow points to the representative flattened, enlarged MES-type cell, the blue arrow points to the representative neuroblast-like cell, and the cyan arrow points to the unhealthy or dead cells indicated by the altered cell morphology, including cellular rounding, shrinkage, and detachment. (F) Heatmaps show the percentage of cell viability after different doses of ACBI1 and etoposide treatment in a panel of NB cell lines ($n = 5$). SK-N-SH cells are treated with the drugs for 3 days and cell proliferation is measured by Incucyte cell confluence assay. The remaining cell lines are treated with the drugs for 4 days, and cell viabilities are measured by CellTiter-Glo Cell Viability Assay. (G) SynergyFinder online tool is used for bliss synergistic analysis to evaluate the synergistic effect of the combination treatment in NB cell lines ($n = 5$). (H) Incucyte cell confluence assays show that the combination of ACBI1 and etoposide (Etop) treatment is more effective in suppressing NB cell proliferation (% confluency) than single agent treatment over time ($n = 5$ wells/group; Error bars indicate SEM). The red arrow is the time point of start to add compounds. Data information: In panel (C) the last time point and panel (D), the data are represented as mean ± SEM. Statistical differences were calculated using ordinary one-way ANOVA. All experiments (A–H) were run at least two independent times, and a representative set is shown. Source data are available online for this figure.

population of flattened monolayer cells persisted in the cultures (Fig. 6E; Appendix Fig. S6B). The ACBI1 and etoposide combination effectively depleted cells with MES-type morphology and hindered the growth of ADRN-type cells (Fig. 6E; Appendix Fig. S6B). We observed that low dose of ACBI1 (40 nM) did not effectively prevent MES-type cells expansion (Appendix Fig. S6C), possibly due to incomplete SWI/SNF ATPases depletion (Appendix Fig. S6A). However, when combined with 500 nM etoposide, this treatment substantially reduced viable cell numbers to a greater extent than the single compound treatment (Appendix Fig. S6C,D). This enhanced effect could be attributed to the reduction of MES signature genes resulting from SWI/SNF ATPases depletion. In light of these findings, it is clear that the combination of ACBI1 and etoposide holds significant potential for effectively treating PDX cells characterized by high plasticity and heterogeneity.

SK-N-SH is a well-established heterogenous NB cell line that gives rise to both ADRN-type and MES-type NB cells. Depleting SWI/SNF ATPases in SK-N-SH cells resulted in a decrease in cell proliferation (Appendix Fig. S6E,F). Consistent with other NB cell lines, RNA-seq data (Dataset EV9) analysis revealed that depleting SWI/SNF ATPases led to negative enrichment of ADRN, MES, EMT, and CCEM signature genes (Appendix Fig. S6G). These results further support a model in which SWI/SNF ATPases depletion reduces the intrinsic plasticity of NB cells and intra-tumor heterogeneity caused by the enrichment of both ADRN and MES signature genes.

Next, we explored the combination of ACBI1 and etoposide treatment in the heterogeneous SK-N-SH cell line, as well as ADRN-type IMR32, BE(2)C, and SY5Y cell lines, to investigate the combination as a potential therapeutic strategy. We observed a variable dose response of these NB cell lines to ACBI1 and etoposide treatment as shown in the heatmaps (Fig. 6F). Importantly, the combination treatment using ACBI1 and etoposide synergistically reduced viable NB cell numbers, with average bliss synergy scores greater than 10 across a range of doses in all tested NB cell lines (Fig. 6G). Incucyte cell confluence assays demonstrated that the individual representative doses of ACBI1 or

etoposide treatment were significantly less effective compared to the combination treatment over time (Fig. 6H). These results support a strategy in which the reduction of MES signature genes after depleting SWI/SNF ATPases sensitizes both heterogenous and ADRN-type NB cells to chemotherapeutic drug treatment.

## Discussion

Neuroblastoma (NB) is a malignancy known for its high plasticity and intra-tumoral heterogeneity, comprising undifferentiated MES-type and predominantly committed ADRN-type cells that can interconvert, which contributes to cancer progression and relapse. In this study, we identified a role for SWI/SNF ATPases in driving transcriptional programs associated with NB cell plasticity and invasiveness. We find that SWI/SNF ATPases enhance plasticity and invasiveness by increasing chromatin accessibility of both ADRN and MES genes in ADRN-type NB cells. Notably, the poised open chromatin of the MES gene, which contains weak enhancers, is primed for activation (Fig. 7). Targeting SWI/SNF ATPases effectively inhibits NB cell proliferation, migration, invasion, and reduces heterogeneity and plasticity (Fig. 7). Moreover, depletion of SWI/SNF ATPases sensitizes NB cells to chemotherapeutic drug treatment. These findings indicate that SWI/SNF ATPases are promising therapeutic targets for high-risk NB patients.

We find that the depletion of SWI/SNF ATPases in NB cells led to reduced chromatin accessibility, diminished enhancer activity, and decreased enhancer-binding of core TFs that drive oncogenic NB biology. These findings align with recent studies emphasizing the importance of SWI/SNF remodeling activity in maintaining nucleosome-free enhancers and facilitating TF binding (Alver et al, 2017; Hargreaves, 2021; Xiao et al, 2022). Core TFs form interconnected feed-forward transcriptional loops in cancers, regulating cell identity and state, which makes them attractive targets for precision therapy (Chen et al, 2020; Saint-Andre et al, 2016). In NB, core TFs play critical roles in determining cell identity, cell viability, cancer progression, and metastasis by

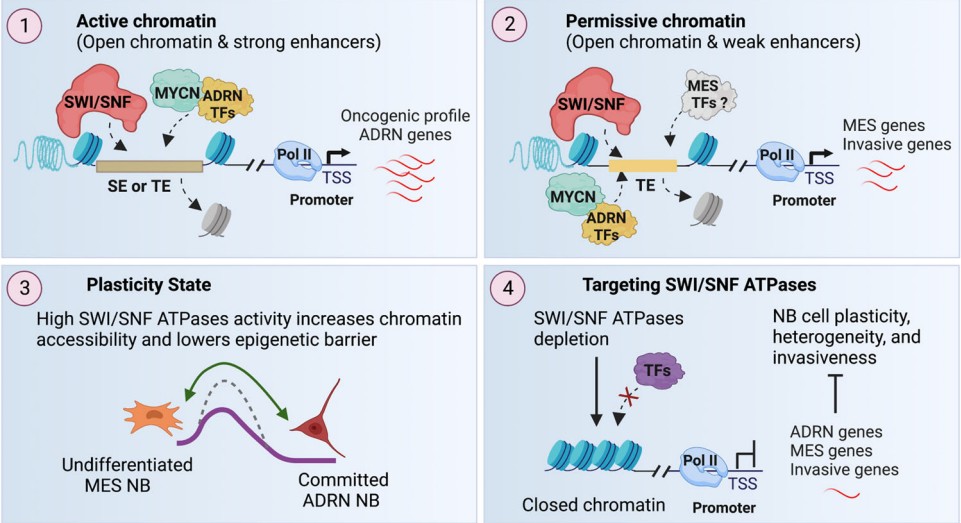

**Figure 7.  Schematic diagram illustrating the impact of SWI/SNF ATPases on ADRN-type NB cells.**

High SWI/SNF ATPases levels increase chromatin accessibility, enhance DNA binding of core TFs, reduce the epigenetic barrier, and promote NB cell plasticity, contributing to intra-tumor heterogeneity, highlighting their potential as appealing therapeutic targets (Created with BioRendere.com). SE super-enhancer, TE typical enhancer.

regulating the expression of genes involved in cell migration, invasion, and extracellular matrix degradation (Boeva et al, 2017; Durbin et al, 2018; Huang and Weiss, 2013; Li et al, 2021; Teitz et al, 2013; van Groningen et al, 2017; Zhu et al, 2017). However, the flexible protein structure of TFs hinders directed drug targeting. Our discovery that targeting druggable SWI/SNF remodeling activity reduces the chromatin binding of core TFs in NB presents a promising strategy to counteract the transcriptional dysregulation and malignant transformation associated with these core TFs. The observed reduction in DNA binding of core oncogenic TFs is not exclusive as ChIP-seq signals for other tested proteins including CTCF, RAD21, and WDR5 at the compacted sites decreased after targeting SWI/SNF ATPases with ACBI1 in IMR32 cells. This observation suggests that the compacted chromatin resulting from the loss of SWI/SNF ATPases acts as a barrier for protein binding. Consistent with our discovery using genetic silencing and PROTAC-mediated protein degradation approaches, a study published during our revisions revealed that treating NB cells with BRM014, an ATPase inhibitor of SMARCA4 and SMARCA2, also reduces the binding of core TFs to genomic loci with decreased chromatin accessibility (Cermakova et al, 2024). This indicates that the ATPase activity of SMARCA2/4 is critical to this process.

Our study shows that SWI/SNF ATPases activity is involved in activating the expression of both ADRN and MES genes in NB, implicating that this activity is crucial in driving transcriptional programs regulating NB cellular plasticity. Permissive chromatin, a crucial regulatory mechanism for precise and cell-specific gene expression, is essential for lineage specification and cancer initiation (Bonifer and Cockerill, 2017; Flavahan et al, 2017; Vicente-Duenas et al, 2018). We discovered that even in ADRN-type of NB, high SWI/SNF ATPases levels maintain a similar chromatin accessibility for both ADRN genes and MES genes. This enables noradrenergic core TFs to not only bind to ADRN genes but also a subset of MES genes, thereby contributing to a basal level

of their expression. These findings suggest that high SWI/SNF ATPases levels in ADRN-type NB establish a permissive chromatin state for MES genes, reducing epigenetic barriers and enabling binding of mesenchymal TFs in response to elevated expression levels triggered by external signals, thereby facilitating trans-differentiation. This aligns with the concept that epigenetic lesions reduce the epigenetic barrier between cell states thereby increasing cellular plasticity and enabling bidirectional transitions (Feinberg et al, 2016; Flavahan et al, 2017).

Some NB models have demonstrated the ability of ADRN and MES cells to transdifferentiate into one another in vitro (van Groningen et al, 2017), while others have shown that ADRN cells can give rise to a heterogeneous population comprising both ADRN and MES cells in vitro, whereas MES cells can only generate heterogeneity in vivo but not in vitro (Thirant et al, 2023). Single-cell data from NB patients revealed that a specific subpopulation of tumor cells in NB corresponds to different stages of SCP/Schwann cell development (Hanemaaijer et al, 2021; Kastriti et al, 2022). Our bulk RNA seq analysis of the SJNBL012407_X1 PDX line showed that when cultured in SCM, cells exhibit features similar to SCPs. This observation suggests that the cell origin of SJNBL012407_X1 might be SCPs, or some SJNBL012407_X1 cells reactivate the developmental gene expression signatures, making them similar to SCPs when cultured in SCM. Furthermore, when these cells are cultured in SCM, akin to SCPs, they exhibit greater plasticity and have the potential to give rise to both ADRN and MES cell types upon switching to FBS-containing media. We find the depletion of SWI/SNF ATPases impairs the ability of PDX cells to adopt a heterogenous morphology during the transition from SCM to FBS-containing media. Our study suggests that targeting SWI/SNF ATPases impedes the SCPs-like cells from giving rise to ADRN cells and MES cells, or suppresses trans-differentiation of ADRN cells into MES cells, as SWI/SNF ATPases are essential for the chromatin accessibility of MES genes in ADRN-type NB. A

limitation of this study is the lack of direct evidence showing the effect of SWI/SNF ATPases on trans-differentiation, which warrants future investigation using single-cell barcoded model systems to confirm this hypothesis.

The depletion of SWI/SNF ATPases hinders SJNBL012407 cells' ability to generate as many ADRN- and MES-like cells when switching cell culture media from SCM to FBS-containing media while promoting neuronal differentiation. These findings are supported by observed cell morphology changes over time (Movies EV1 and EV2; Fig. 5A; Appendix Fig. S5C), bulk RNA-seq data (Appendix Fig. S5K,M), and single-cell RNA-seq results (Fig. 5D,E; Appendix Fig. S5N,O). These findings, in combination with the Incucyte cell cytotoxicity assay (Appendix Fig. S5E,F), are consistent with a model that the depletion of SWI/SNF ATPases impairs the ability of PDX cells to adopt a heterogenous morphology during the transition from SCM to FBS-containing media, and this impairment may be caused by a combination effect involving the suppression of cell proliferation, induction of cell death, and reduction in the ADRN and MES transcriptional profiles. Additionally, bulk-RNA seq results indicated that depletion of SWI/SNF ATPases resulted in a reduction of both ADRN and MES signatures in ADRN-type NB cell lines, the heterogenous NB cell line SK-N-SH, and heterogenous NB PDX cells. These findings are particularly important, considering that MES cells are enriched after chemotherapy treatment of NB cells and in tumors from relapsed patients (Boeva et al, 2017; Gartlgruber et al, 2021; Gautier et al, 2021; Ponzoni et al, 2022; van Groningen et al, 2017). Indeed, our results demonstrate that the reduction of MES signatures in NB cells after depleting SWI/SNF ATPases sensitizes both heterogeneous and ADRN-type NB cell lines to chemotherapeutic drug treatment.

The dual SMARCA2/4 inhibitors showed certain toxicity in vivo (Papillon et al, 2018). This toxicity could be due to targeting SMARCA2/4 in normal tissues in mice or because of targeting other proteins. Interestingly, unlike the dual SMARCA2/4 inhibitors, prolonged treatments with the SMARCA2/4 PROTAC degrader AU-15330 showed no evident toxicity in immune-competent mice (Xiao et al, 2022). This suggests that the SMARCA2/4 PROTAC degraders are less toxic than the dual SMARCA2/4 inhibitors in mice, although we cannot rule out the potential off-target effects in patients. Since SMARCA2/4 PROTAC degraders and chemotherapeutic drugs have a synergistic effect on suppressing NB growth, utilizing low doses of SWI/SNF ATPases degraders in this context could potentially minimize the off-target effects associated with targeting SWI/SNF ATPases in vivo.

In conclusion, our findings contribute to the growing body of research highlighting the critical role of SWI/SNF chromatin remodeling activity in driving cancer progression. We provide mechanistic insights into how SWI/SNF ATPases promote the binding of core TFs to DNA, shedding light on the underlying molecular processes involved in core TFs-driven NB progression. Moreover, our study provides mechanistic insights into the heterogenous nature of NB. These results highlight the potential of targeting SWI/SNF ATPases as an innovative therapeutic strategy for high-risk NB patients with core TF dependencies. Depmap Predictability function analysis indicates that NB cell lines with lower expression of *SMARCA2* tend to be more sensitive to the loss of *SMARCA4* (Appendix Fig. S1G), which suggests that targeting both SMARCA2 and SMARCA4 with PROTAC degraders

is a promising approach in treating NB patients. Our discoveries pave the way for further research and potential therapeutic interventions to address NB chemoresistance, tumor heterogeneity, and relapse in high-risk NB.

# Methods

## Cell culture

Human neuroblastoma (NB) cell lines IMR32, IMR5, SK-N-BE(2)C (BE(2)C), SMS-KCNR (KCNR), SH-SY5Y (SY5Y), and SK-N-SH were obtained from the cell line bank of the Pediatric Oncology Branch of the National Cancer Institute and have been genetically verified. IMR5 is a subclone of IMR32 but exhibits different cell morphology, adhesive ability, and proliferation rate. ARPE-19 is a spontaneously arising human retinal pigment epithelial (RPE) cell line obtained from ATCC (cat # CRL-2302). All the above NB and ARPE-19 cell lines were maintained in RPMI-1640 complete media supplemented with 10% fetal calf serum (FBS), 100 µg/mL streptomycin, 100 U/mL penicillin, and 2 mM L-glutamine. Cells were grown at 37 °C with 5% $CO_2$. We generated IMR5, IMR32 and BE(2)C stably luciferase-GFP-expressing cells by lentiviral transduction and subsequent selection with 0.5 µg/mL of puromycin. PDX cell lines SJNBL012407_X1 (MAST97) was obtained from St Jude Children's Research Hospital. SJNBL012407_X1 expanded from orthotopic sites of xenografts were first cultured in neural stem cell media: DMEM/F12 + GluMax containing EGF (20 ng/ml), basic FGF (40 ng/ml), B-27 supplement and streptomycin/penicillin. To treat SJNBL012407_X1 cells with compound, the cells were cultured in RPMI-1640 complete media overnight on matrigel coated plates before drug treatment and with cell cultures maintained in complete media. All the cell lines were tested mycoplasma contamination-free. Genetic mutations of key oncogenes of NB cell lines used in this study were listed below:

SY5Y: SMARCA4 (R973W) and ALK (F1174L) indicated by depmap portal (https://depmap.org/portal/). ALK mutation in SY5Y was also reported by George et al (2008).

SK-N-SH: SMARCA4 (R973W) and ALK (F1174L) indicated by depmap portal (https://depmap.org/portal/).

BE(2)C: TP53 (C135F) (Van Maerken et al, 2006) and NF1 (loss of heterozygosity) (Holzel et al, 2010), MYCN amplification.

KCNR: ALK (R1275Q) (George et al, 2008), MYCN amplification.

IMR32: NF1 (R69G) and CCND2 (L214I) indicated by depmap portal (https://depmap.org/portal/), MYCN amplification.

IMR5: MTOR (F1888V), CCND2 (L214I) indicated by depmap portal (https://depmap.org/portal/), MYCN amplification.

## Transient transfection

Transient transfection was performed as described previously (Liu et al, 2006). Control siRNA (siCtrl, Cat. # D-001810-01) and siRNAs targeting different genes (*siSMARCA4_5*, Cat. # J-010431-05; *siSMARCA4_7*, Cat. # J-010431-07; and *siSMARCA2_1*, Cat. # D-017253-01) were purchased from GE Dharmacon. siRNAs were transiently transfected into NB cells using Nucleofector electroporation (Lonza): solution L and program C-005 for IMR32 and IMR5, and solution V and program A-030 for KCNR.

     

## Two-dimensional (2D) cell proliferation and viability assay

To evaluate cell proliferation, IMR5 ($1.2 \times 10^4$), BE(2)C ($1 \times 10^4$), ARPE ($0.4 \times 10^4$), IMR32 ($2 \times 10^4$), KCNR ($1.5 \times 10^4$), SY5Y ($1.5 \times 10^4$) cells were seeded into regular flat bottom cell culture grade 96-well plates and incubated overnight. The next day, cells were treated with different doses of SMARCA2/4 dual inhibitors ACBI1 (Selleckchem, Catalog No. S9612) and AU-15330 (Selleckchem, Catalog No. E1103), and a DMSO control. To determine the combined effect of ACBI1 and etoposide on NB cells, SK-N-SH ($1.5 \times 10^3$), BE(2)C ($1.5 \times 10^3$), IMR32 ($3.2 \times 10^3$), SY5Y ($4 \times 10^3$) cells were seeded into flat bottom cell culture grade 384-well plates. After overnight incubation, cells were treated with varying concentrations of ACBI1 and etoposide. The growth kinetics of one plate were monitored through the Incucyte ZOOM or SX5 (Sartorius) using the integrated confluence algorithm as a surrogate for cell number. Four days after treatment, a CellTiter-Glo® luminescent assay (Promega, catalog number G9242) was performed on the other plate. Cell viabilities of DMSO-treated cells were set to 100% and IC-50 curves were generated with the GraphPad Prism (RRID:SCR_002798) software. A representative experiment has been repeated at least two to three times. To investigate the impact of ACBI1 and AU-15330 on NB cell apoptosis, we treated IMR5, IMR32, and BE(2)C cell lines with varying doses of ACBI1 or AU-15330. Apoptosis was evaluated using an Incucyte Caspase-3/7 green dye according to the manufacturer's instructions (Sartorius, catalog number 4440). The Incucyte SX5 (Sartorius) captures cell images, and the apoptotic cells were quantified using Incucyte image analysis tools. To investigate the cytotoxic effect of ACBI1 on SJNBL012407_X1 PDX cells, we used the Incucyte Cytotox Green Dye (Sartorius, catalog number 4633) cell cytotoxicity assay, which assesses cell membrane integrity. Cytotoxic cell death was quantified using Incucyte image analysis tools.

To determine the duration of effectiveness for the ACBI1 and AU-15330 drugs in 2D culture media, or indirectly measure the half-life of ACBI1/AU-15330, IMR32 cells were treated once with varying doses of ACBI1 and AU-15330 and cultured for up to 6 days. Subsequently, a western blot was performed to evaluate SMARCA2/4 protein levels at each time point.

## Three-dimensional (3D) spheroid proliferation assay

Cells (2000–4000) were plated into each well of a 96-well round bottom, ultra-low attachment plate (Corning Cat. 7007), and the plates were centrifuged for 10 min at $125 \times g$ at room temperature. To allow for spheroid formation, the plates were incubated for 48 h before treating with DMSO, ACBI1, and AU-15330. Two days following the initial treatment, cells were treated again with DMSO, ACBI1, and AU-15330, and the plates were centrifuged for $300 \times g$ at 5 min. Two days following the previous treatment, the cells were then treated one more time with DMSO, ACBI1, and AU-15330 (treatments on day 2, day 4, and day 6 post-initial plating). Spheroid areas were monitored through the Incucyte Spheroid software (Sartorius). Note: some of the spheroids were not captured by the camera of the Incucyte machine at certain time points possibly due to their inappropriate position in the wells and the small size of these spheroids. Consequently, these wells were excluded from the analysis.

## Cell scratch wound assay

For scratch wound assay, Essen ImageLock 24-well plates were first coated with Collagen (Sigma, 3867). Then $2.1 \times 10^5$ BE(2)C cells or $2.8 \times 10^5$ IMR5 cells were plated into each well of the 24-well plate. The next day, the cells were treated with or without ACBI1 or AU-15330. 24 h later, the cells were incubated with 0.1 µg/ml (for IMR32) or 12 µg/ml (for BE(2)C) mitomycin-C (Sigma: M5353) for 2 h to inhibit cell proliferation. Following the 2 h incubation, cells were scratched using the Essen 24-well WoundMaker (Ann Arbor, MI, USA) following their protocols, and the media was changed to RPMI with or without ACBI1 and AU-15330. The wound confluence was obtained and analyzed by using the IncuCyte phase-contrast imaging and scratch wound assay system and software (Sartorius).

## 3D spheroid cell invasion assay

3D Spheroid Invasion Assays were conducted following the Sartorius protocol (IncuCyte® 3D Spheroid Invasion Assay). 2000 IMR32-luc-GFP, BE(2)C-luc-GFP, or 4000 IMR32-luc-GFP cells were plated into each well of a 96-well round bottom, ultra-low attachment plate (Corning Cat. 7007). Plates were then centrifuged for 10 min at $125 \times g$ at room temperature. To allow for spheroid formation, the plates were incubated for 48 h before treating with DMSO, ACBI1, and AU-15330. Two days following initial treatment, Matrigel (Corning, Cat. 354234) was added at a final concentration of 1.5 mg/mL for IMR5 and BE(2)C, 1.1 mg/mL for IMR32 cells, and the plates were centrifuged at $300 \times g$ for 5 min at 4 °C. After ensuring that the spheres were still located in the center, the plates were then incubated at 37 °C for 30 min to allow for polymerization of the Matrigel. After polymerization, media was added containing DMSO, ACBI1, and AU-15330. AU-15330 was applied every 48 h thereafter. The plates were monitored using the IncuCyte Single Spheroid Invasion software (Sartorius). For 3D matrigel spheroid cultures, the impact of ACBI1 and AU-15330 on NB cell death was assessed using the LDH-Glo Cytotoxicity Assay following the manufacturer's instructions (Promega, catalog number J2380). Note: some of the spheroids were not captured by the camera of the Incucyte machine at certain time points possibly due to their inappropriate position in the wells and the small size of these spheroids. Consequently, these wells were excluded from the analysis.

## Protein isolation and western blotting analysis

For assessment of protein levels, cells were lysed using RIPA buffer, and 10 µg of total protein was separated and electroblotted as described previously (Liu et al, 2017). Protein bands probed with diluted primary antibodies (Dataset EV10) were detected using a goat anti-rabbit or mouse IgG-HRP conjugated secondary antibody (200 µg/mL; Santa Cruz Biotechnology) and visualized using enhanced chemiluminescence.

## RNA-seq

Total RNA isolated from NB cells with different treatments were subjected to RNA-seq analysis as previously described (Liu et al, 2020b). Total RNA was extracted using the RNeasy Plus Mini Kit (Qiagen Inc.) according to the manufacturer's instructions. TruSeq® Stranded Total RNA LT Library Prep Kit or TruSeq

Stranded mRNA Library Prep kit (Illumina, San Diego, CA, USA) was used for preparing Strand-specific whole transcriptome sequencing libraries by following the manufacturer's procedure. RNA-seq libraries were sequenced by paried-end on Illumina sequencer. The Fastq files with paired-end reads were processed using Partek Flow. The raw reads are aligned using STAR and the aligned reads are quantified to annotation model through Partek E/M. The normalization method used here is counts per million (CPM) through Partek Flow. Gene set enrichment analysis (GSEA) was further used for data analysis. By default, the false discovery rate (FDR) less than 0.25 is significant in GSEA.

## ATAC-seq

ATAC-seq was performed using the ATAC-seq kit (Active Motif, cat. 53150) following the manual, and the data analysis was performed as previously described (Liu et al, 2020b; Xu et al, 2023). Approximately, 100,000 NB cells were pelleted and the Tn5 transposition reaction was performed with the Nextera kit (Illumina), according to the manufacturer's protocol. ATAC libraries were sequenced on an Illumina NextSeq 2000 machine (paired-end 100-bp reads). The Fastq files were processed using Encode ATAC_DNase_pipelines (https://github.com/kundajelab/atac_dnase_pipelines) installed on the NIH biowulf cluster (https://hpc.nih.gov/apps/atac_dnase_pipelines.html). Biological duplicates in each condition were pooled together for the downstream analysis.

The peak sets were further analyzed using the deepTools2 suite (v3.3.0) (Ramirez et al, 2016). By using bamCoverage, peaks were normalized to reads per kilobase per million reads normalized read numbers (RPKM). ComputeMatrix function of the deepTools was used to generate a matrix of signal intensity of ATAC-seq peak centers (±500 bp, total 1000 bp), as intensity scores in 10 bp bins. The matrix of signal intensity was further used to calculate the accumulated signal around each peak center to be used as signal intensity for ATAC-seq peaks. Peaks in the control cells with a signal intensity (read counts) greater than 300 are deemed significant and are chosen for downstream analysis. To determine differential ATAC-seq peaks between control and treatment group, a 2.5-fold change of ATAC-seq signal intensities was used as cutoff for IMR32 and BE(2)C cells (ACBI1 vs. DMSO), and a 2-fold change of ATAC-seq signal intensities was used as cutoff for IMR5 cells (for both siSMARCA4_5 vs. siCtrl and siSMARCA4_7 vs. siCtrl).

## ChIP-seq

ChIP-seq was performed using the ChIP-IT High Sensitivity kit (Active Motif, cat. 53040) as described previously (Liu et al, 2020b). Briefly, formaldehyde (1%, 13 min) fixed cells were sheared to achieve chromatin fragmented to a range of 200–700 bp using an Active Motif EpiShear Probe Sonicator. IMR32 cells were sonicated at 25% amplitude, pulse for 20 s on and 30 s off for a total sonication "on" time of 16 min. Sheared chromatin samples were immunoprecipitated overnight at 4 °C with antibodies targeting H3K27ac, MYCN, HAND2, PHOX2B, GATA3, RAD21, CTCF, and WDR5 (Dataset EV10). Active Motif ChIP-seq spike in chromatin (Active Motif Cat No. 53083) and *Drosophila* specific histone variant H2Ac (Active Motif Cat No. 61686) were included in the ChIP experiments. Libraries prepared with NEBNext Ultra II

DNA Library Prep Kit (NEB, E7645) were multiplexed and sequenced on an Illumina NextSeq machine.

As described previously (Liu et al, 2020b), for the home generated ChIP-seq data, ChIP enriched DNA reads were mapped to reference human genome (version hg19) using BWA. ChIP-seq read density values were normalized per million mapped reads. High-confidence ChIP-seq peaks were called by MACS2 (https://github.com/taoliu/MACS) with the narrow peak for each TF. Peaks from ChIP-seq of SMARCA4 and CRC TFs were selected based on $p$-value ($p < 10^{-5}$ for MYCN, GATA3 and PHOX2B, $p < 10^{-7}$ HAND2 and H3K27ac). The distribution of peaks (as intronic, intergenic, exonic, etc.) was annotated using HOMER. Enrichment of known and de novo motifs were found using HOMER script "find Motifs Genome.pl" (http://homer.ucsd.edu/homer/ngs/peakMotifs.html).

The peak sets were further analyzed using the deepTools2 suite (v3.3.0) (Ramirez et al, 2016). By using bamCoverage, peaks were normalized to reads per kilobase per million reads normalized read numbers. ComputeMatrix function of the deepTools was used to generate a matrix of signal intensity of TFs of their peak centers (±500 bp, total 1000 bp), as intensity scores in 10 bp bins. The matrix of signal intensity was further used to calculate the accumulated signal around each peak center to be used as signal intensity for each H3K27ac and TF binding sites.

Metagene plots of signal intensity of ChIP samples were generated using deepTools. Briefly, computeMatrix was used to calculate signal intensity scores per ChIP sample in a given genome region that was specified by a bed file. The output of computeMatrix was a matrix file of scores of two ChIP samples which was then used to generate the heatmap using plotHeatmap function, generate composite plot using plotProfile function.

## Single-cell RNA-sequencing (scRNA-seq)

NB PDX cells maintained at different culturing media with or without ACBI1 treatment were harvested for scRNA-seq analysis. The samples in single-cell suspension were captured using the Chromium Controller (10x Genomics) and subjected to the 10x Genomics 3' v3.1 chemistry kit for library generation according to the manufacturer's protocol. The cDNA libraries were sequenced on an Illumina NovaSeq 6000 sequencer with a targeted depth of ~40,000 reads per cell. The scRNA-seq results were demultiplexed and processed with the Cell Ranger pipeline offered by the 10X Genomics to generate count matrices for downstream analysis.

The count matrices from each sample were first corrected in SoupX (Young and Behjati, 2020) to eliminate the ambient RNA contamination using a contamination rate of 0.2. Next, the corrected count matrices were analyzed in scrublet (Wolock et al, 2019) to predict doublets with the estimated doublet rate set at 0.008 per thousand cells captured in each capture lane before the data were further analyzed in R package Seurat (v4.3.0) (Butler et al, 2018). In Seurat, low-quality cells (cells with less than 300 or more than 7500 genes detected, doublets, cells with more than 70% mitochondria content, and cells with less than 500 or more than 50,000 unique molecular identifiers detected) were first excluded. The remaining 10488 cells were next normalized and scaled to regress out the variabilities caused by mitochondrial gene content, ribosomal gene content, number of RNA molecules detected, number of genes detected, and cell cycle stage in the cells. Furthermore, reciprocal PCA (rPCA) was applied to integrate cells from the three capture lanes to minimize the influence of batch effect on the downstream

comparisons. Finally, gene lists of MES and ADRN were derived from previous report (van Groningen et al, 2017) and AddModuleScore function in Seurat was used to calculate the average expression of genes involved in each gene list in all 10488 cells.

## Statistics and reproducibility

The statistical analyses used throughout this paper are specified in the appropriate results paragraphs and Methods sections. Additional statistical analyses were performed using standard two-tailed Student's *t*-test, one-way ANOVA, and the software GraphPad Prism 8.1.0.

## Data availability

All the home generated ChIP-seq, ATAC-seq, bulk RNA-seq and single-cell RNA-seq can be found in the Gene Expression Omnibus (GEO) database. GEO accession number for ChIP-seq, ATAC-seq and bulk RNA-seq data generated in this study is GSE240593. The demultiplexed raw sequencing data and processed count matrices for scRNA-seq data were deposited to GEO under the accession number GSE240724. RNA-seq data of *HAND2* and *MYCN* silencing for 72 h in IMR32 can be found under GEO accession number GSE183641.

The source data of this paper are collected in the following database record: biostudies:S-SCDT-10_1038-S44318-024-00206-1.

## Peer review information

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

## Acknowledgements

We thank Drs. Madeline Wong, Liz Conner, and Mr. Steven Shema from NCI Genomics Core for DNA and RNA sequencing. We thank Drs. Michael Kelly, Kimia Dadkhah, and Anna Lee Fong from NCI Single-Cell Analysis Facility for single-cell RNA sequencing. We thank Dr. Jeyshka Reyes Gonzalez from POB, NCI for her assistance. This work utilized the computational resources of the NIH HPC Biowulf cluster (http://hpc.nih.gov). Part of the figures were created with BioRender.com. This work was funded by the Center for Cancer Research, Intramural Research Program at the National Cancer Institute. Grant number: Z01:ZIA BC 010788; PI: Carol J Thiele.

## Author contributions

**Man Xu**: Conceptualization; Data curation; Formal analysis; Validation; Investigation; Visualization; Methodology; Writing—review and editing. **Jason J Hong**: Data curation; Formal analysis; Validation; Investigation; Visualization; Methodology; Writing—review and editing. **Xiyuan Zhang**: Data curation; Formal analysis; Investigation; Visualization; Methodology; Writing—review and editing. **Ming Sun**: Investigation; Methodology; Writing—review and editing. **Xingyu Liu**: Data curation; Formal analysis; Validation; Investigation; Visualization; Methodology; Writing—review and editing. **Jeeyoun Kang**: Data curation; Investigation; Methodology; Writing—review and editing. **Hannah Stack**: Data curation; Investigation; Methodology; Writing—review and editing. **Wendy Fang**: Data curation; Investigation; Methodology; Writing—review and editing. **Haiyan Lei**: Data curation; Formal analysis; Validation; Investigation; Methodology; Writing—review and editing. **Xavier Lacoste**: Data curation; Formal analysis; Validation; Investigation; Visualization; Methodology; Writing—review and editing. **Reona Okada**: Investigation; Methodology; Writing—review and editing. **Raina Jung**: Investigation; Methodology; Writing—review and editing. **Rosa Nguyen**: Investigation; Methodology; Writing—review and editing. **Jack F Shern**: Investigation; Methodology; Writing—review and editing. **Carol J Thiele**: Conceptualization; Resources; Supervision; Funding acquisition; Investigation; Methodology; Project administration; Writing—review and editing. **Zhihui Liu**: Conceptualization; Data curation; Formal analysis; Supervision; Validation; Investigation; Visualization; Methodology; Writing—original draft; Project administration.

Source data underlying figure panels in this paper may have individual authorship assigned. Where available, figure panel/source data authorship is listed in the following database record: biostudies:S-SCDT-10_1038-S44318-024-00206-1.

## Funding

## Disclosure and competing interests statement

The authors declare no competing interests.

