## [Peer Review File · The EMBO Journal]

Targeting SWI/SNF ATPases reduces neuroblastoma cell plasticity

Man Xu, Jason Hong, Xiyuan Zhang, Ming Sun, Xingyu Liu, Jeeyoun Kang, Hannah Stack, Wendy Fang, Haiyan Lei, Xavier Lacoste, Reona Okada, Raina Jung, Rosa Nguyen, Jack Shern, Carol Thiele, and Zihui Liu

Corresponding author(s): Zihui Liu (liuzhihu@mail.nih.gov) , Carol Thiele (thielec@mail.nih.gov)

Review Timeline:

Submission Date:	26th Sep 23
Editorial Decision:	25th Nov 23
Revision Received:	10th May 24
Editorial Decision:	25th Jun 24
Revision Received:	1st Jul 24
Accepted:	17th Jul 24

Editor: Cornelius Schneider

Transaction Report:

Dear Dr. Liu,

Thank you for submitting your manuscript for consideration by the EMBO Journal. We have now received comments from three reviewers, which are included below for your information.

As can be seen from the reports, all three referees found the results of importance and interest, and agree that the experiments were performed competently. I find the referee comments balanced and fair and would therefore invite you to address them in a revised version of the manuscript.

I feel that the technical concerns voiced by referee #1 are important to would make major parts of the manuscript more rigorous and well-rounded. Regarding the comments raised by referee #3 I would not require additional experiments but rather ask for a more balanced discussion that would mention some of the challenges associated with targeting these proteins. I am happy to to discuss the revision in more detail via email or phone/videoconferencing if you have any additional questions. I should also add that it is The EMBO Journal policy to allow only a single major round of revision and that it is therefore important to resolve the main concerns at this stage.

We generally allow three months as standard revision time, which can be extended to 6 months in case of major revisions, such as the experiments required here. As a matter of policy, competing manuscripts published during this period will not negatively impact on our assessment of the conceptual advance presented by your study. However, we request that you contact the editor as soon as possible upon publication of any related work, to discuss how to proceed. Should you foresee a problem in meeting the deadline, please let us know in advance and we may be able to grant an extension.

Thank you for the opportunity to consider your work for publication. I look forward to your revision.

Yours sincerely,

Cornelius Schneider

Cornelius Schneider, PhD
Editor
The EMBO Journal
c.schneider@embojournal.org

- a point-by-point response to the referees' comments, with a detailed description of the changes made (as a word file).
- a word file of the manuscript text.
- individual production quality figure files (one file per figure)
- a complete author checklist, which you can download from our author guidelines

(<https://www.embopress.org/page/journal/14602075/authorguide>).
- Expanded View files (replacing Supplementary Information)
Please see out instructions to authors
<https://www.embopress.org/page/journal/14602075/authorguide#expandedview>

We realize that it is difficult to revise to a specific deadline. In the interest of protecting the conceptual advance provided by the work, we recommend a revision within 3 months (23rd Feb 2024). Please discuss the revision progress ahead of this time with the editor if you require more time to complete the revisions. Use the link below to submit your revision:

Referee #1:

EMBOJ-2023-115707

General summary and opinion about the principle significance of the study, its questions and findings:

The manuscript titled "Targeting SWI/SNF ATPases to Counteract Neuroblastoma Lineage Plasticity and Chemotherapy Resistance" presents an informative investigation into the role of the SWI/SNF chromatin remodeling complex in Neuroblastoma (NB), providing valuable insights into its implications for tumor cell heterogeneity and therapy responsiveness. Xu et al. effectively demonstrate that targeting SWI/SNF ATPases, particularly utilizing SMARCA2/4 dual degraders, leads to significant inhibition of key cellular processes in NB, such as cell proliferation and invasion, while also exerting a pronounced impact on cellular plasticity. The manuscript highlights these outcomes as central findings with significant therapeutic potential. However, there are concerns regarding the interpretation of the presented experimental results in relation to effects on cell proliferation of the dual degraders. To ensure the manuscript's readiness for potential publication in EMBO Journal, the authors are encouraged to address these concerns by assessing both the proliferative and apoptotic statuses of cells subjected to treatment over a four-day duration with SMARCA2/4 inhibitors. These additional experiments will significantly strengthen the manuscript and contribute to a more robust and comprehensive understanding of the presented findings.

Specific major concerns essential to be addressed to support the conclusions:

- In Figure 2 Xu et.al. present a robust effect on chromatin accessibility and decreased global binding of several transcription factors known to promote an ADRN phenotype in NB cells. It appears that the reduced chromatin accessibility observed with ACB1 and siSMARCA4 treatments is a global event, which raises the question if not nearly all transcription factors will have reduced average binding to DNA? To address this question authors could add additional CHIP data on other transcription factors that are not affected by SMARCA2/4 inhibition. If nearly all transcription factors have less binding to DNA, it is important to rewrite parts of the discussion and highlight that the observed reduction of DNA binding of core oncogenic transcription factors is not an exclusive event.
- In Figure 3 authors show that ACB1 and AU-15330 treatment reduce invasion of NB cells. However, in Figure 1 authors also show that the same concentrations of these inhibitors lead to less cell proliferation. Further experiments are therefore needed to evaluate if impaired invasion is simply because cells are damaged or dying. By monitoring cell proliferation more carefully and staining for apoptotic markers this issue could be addressed. Claiming that GFP expression is indicative of cells being healthy is not sufficient evidence.
- Similar to concerns regarding Figure 3, it is not clear how to interpret results in Figure 5. Given that the ACB1 affects cell proliferation it is not clear whether cellular clones are selected for due to toxicity or if they trans-differentiate. As mentioned in the discussion a barcode model would be preferable to further study these possibilities, however it would be adequate to provide controls for cell proliferation and cell death in the current experimental set up. Extensive toxicity could indicate that cellular clones are being selected for in this experimental set up.

Minor concerns that should be addressed:

- In Figure 1d what is the rationale to use retinal pigment epithelia (RPE) cell line as a control for neuroblastoma cell lines? Also, does ACBI1 lead to SMARCA2/SMARCA4 degradation in ARPE19?
- In Figure 1e it is not clear that the degradation is time dependent. More time points should be added, and quantifications of the western blot bands are required to draw that conclusion.
- In Figure 3 ACBI1 is used at a range of different concentrations from 100nM to 1000nM. Is this a typo? If not, why do different cells require such different concentrations - the amount of SMARCA4 in these cells (Fig 1d) is rather similar.
- Similarly in Figure 3d, AU-15330 (200nM or 2000nM) does not seem consistent in the quantified result and image.
- In Figure 3, in the 3D spheroid cell invasion assay, cells were only treated cell once. What is the half-life of SCBI1/AU-15330?
- In Figure 5a there is no visual difference (to the human eye) between the DMSO and ACBI1 conditions and authors should present quantifications of cellular morphologies in main Figure 5.
- In Figure 6d quantifications of cell morphology are needed and statistical analysis to determine differences between treatments.
- In Figure 6g, something looks strange with the SY5Y graph, since the assays were started with similar amounts of cells according to the materials and methods.
- The authors should label key oncogene mutation status in the NB cell lines they are using. For example, SH-SY5Y harbors an SMARCA4-R973W mutation. (https://depmap.org/portal/cell_line/ACH-001188?tab=overview)
- In ref 46, Jubierre et al. used the same cell line (BE2(C)) and found knockdown of SMARCA4 induced caspase-dependent apoptosis. Do the authors observe apoptosis in their system?
(46. Jubierre, L. et al. BRG1/SMARCA4 is essential for neuroblastoma cell viability through modulation of cell death and survival pathways. *Oncogene* 35, 5179-90 (2016).
- Figure 1, the data from DepMap portal could be updated to 23Q2 which includes more NB cell lines. DepMap Public 23Q2+Score, Chronos (n=33) and Expression Public 23Q2 (n=37)

Referee #2:

Neuroblastoma disproportionally accounts for 15% cancer related deaths in children. With intensive multimodal therapies, only 50% high-risk patients get cured. While a number of reasons could contribute to the failure of current therapies, it is believed that cellular plasticity of neuroblastoma may play an important role in chemoresistance. It has been long noticed that neuroblastoma cells are a heterogenous population by Dr. Thiele, the senior author of this paper. Recent studies by using RNA-seq and epigenetic approaches confirmed the heterogeneity of neuroblastoma cells at molecular levels, leading to a definition of ADRN and MES types of neuroblastoma cells. While the literature proposed that ADRN and MES are interconvertible, it remains puzzling that there are not much solid evidence showing the MES neuroblastoma cells are pre-existing in tumors, which are believed to be responsible for chemoresistance. In this elegant study, Drs. Thiele and Liu group made a further contribution by showing that SWI/SNF ATPases are essential in establishing a MES-permissive chromatin state in ADRN neuroblastoma, promoting cell lineage plasticity. By applying ATAC-seq, ChIP-seq and RNA-seq in combination with genetic and pharmacologic degradation of SMARCA4 and SMARCA2, they clearly showed that SWI/SNF ATPases regulate the chromatin accessibility for both ADRN and MES genes in the so called ADRN cell lines. Importantly, they found that SWI/SNF ATPases are important to tumor cell invasion, and the PROTAC degrader against SMARCA4/SMARCA2 suppressed both cell invasion and cell proliferation. Further, they demonstrated that the SMARCA4/SMARCA2 degrader significantly enhanced the anticancer effect of etoposide. Their findings not only provide an insight into understanding the mechanism of neuroblastoma cell plasticity but also a good rationale to target SWI/SNF ATPases to overcome the chemoresistance conferred by MES state. Overall, I really enjoyed reading this MS and recommend publication in EMBO J after clarification.

I have some comments for the authors to consider clarifying .

1. The authors should not overlook the importance of SMARCA2. The DepMap results clearly show that high levels of SMARCA2 is negatively correlated the knockout effect of SMARCA4 (Predictability function in DepMAP after searching SMARCA4). I would suggest the authors add this data into their manuscript, and change tone to address the potential effect of SMARCA2. Also, the

- presence of SMARCA2 will certainly affect the PROTAC effect against SMARCA4 by titrating the drug. The authors should acknowledge this. The current PROTAC targeting both is not a bad idea at all.
2. The authors claimed APRE-19 cell as a normal cell. Actually, no cells are normal once they become immortalized. "Non-cancerous" cell might be more appropriate. It would be helpful for readers to understand the cell of origin of APRE-19 and why the authors chose this cell line as a control.
 3. In their 3D spheroid experiment, different cell lines were treated with different concentrations of drug. Can they justify this? Can they discuss what potential factors led to the differential responses of neuroblastoma cell lines to the drug treatment?
 4. In Fig. 2f the authors argued that the effect ChIP-seq signals of H3K27ac and other TFs by siSMARCA4 less than the ACBI1 treatment may be due to lower effect of knockdown, which might be true; however, the authors should discuss the compensatory role of SMARCA2 in IMR32 cells since the western blot showed high levels of expression of SMARCA2.
 5. In Figure 2h, GSEA results showed a bimodal pattern. Can the authors explain why?
 6. It is very interesting that bulk-RNA-seq identified SCP signature in PDX models when cultured in SCM but shift to ADRN/MES under FBX media. Based on recent scRNA-seq studies to understand the cell of origin of neuroblastoma cells, can author give a further discussion how their findings are linked to the neuroblastoma origin? In Fig. 5c,d, ACBI1 treatment of FBS cells pushed the ADRN/MES score to a similar degree as in SCM. Does this mean inhibition of SMARCA 4/2 in neuroblastoma cells lead to a more stem cell like state (although not necessarily mean it will increase therapy resistance)? Have they looked into the effect of ACBI1 on PDX cells cultured in SCM?
 7. In their scRNA-seq analysis, could the authors give details about the features of each cluster with or without ACBI1 treatment?
 8. Lastly, the authors should acknowledge that the phenotype they observed was through genetic depletion and PROTRAC-mediated protein degradation. Whether the ATPase activity of SMARCA4/2 is critical to what they observed remains to be studied.
 9. One caveat of this study is lack of in vivo test probably due to the issue of PROTAC.

Referee #3:

This manuscript describes the effects of applying recently developed degraders of SMARCA2 and SMARCA4 proteins to NB cells. The compounds cause growth defects in NB cells. To further investigate the underlying mechanisms changes to chromatin accessibility, transcription factor binding and transcription are measured. Loss of transcription factor binding is correlated with downregulation of genes linked to the invasiveness of NB cells. In addition, ACBI1 treatment reduces invasiveness. Similarly, genes linked to the adrenal and mesenchymal subtypes of NB are reduced following ACBI1 treatment suggesting a role for the ATPases in epigenetic plasticity. It is also observed that combined ACBI1 treatment results in a synergistic reduction in cell numbers. The findings indicate how NB cells respond to loss of the SMARCA ATPases. This will be of interest to researchers studying NB.

The manuscript also mentions the potential for a therapeutic value of SMARCA4 in NB. SMARCA4 has a net negative effect across all cell lines (depmap), its essential for mouse development, loss of function mutations are associated with diverse cancers, so there is a likelihood of side effects when using SMARCA4 inhibitors therapeutically. Indeed, in a human lung xenograft model, toxicity and body weight loss limited therapeutic effects <https://pubs.acs.org/doi/10.1021/acs.jmedchem.8b01318>. Although, NB cells may be more dependent on SMARCA4 than some other cell types there will still be significant potential for off target effects. In order to justify the therapeutic angle presented in the manuscript more information should be provided to identify contexts in which the benefits are likely to outweigh deleterious off target effects.

The data presented appear to be of high quality. The following minor points were noted.

What proportion of accessible chromatin peaks are unaffected by ACBI1?

CHIP for various factor is shown in figure 2e and f at sites where ATAC-seq is changing. Are changes at these sites representative of the genome wide distribution of these factors? If these factors are also bound at sites where ATAC does not change the effects may be smaller. Data indicating how ChIP changes at sites where chromatin accessibility does not change should also be shown.

The effect on ACBI1 on transcription factor occupancy is not uniform. Stronger effects are observed on HAND2 and K27ac, minor effects on MYCN and GAA3. This should be commented. On.

The evidence that SMARCA2 plays a limited role in NB is limited. The combined effect of siSMARCA4/2 is not compared to siSMARCA4 alone.

The transcriptome analysis in Figure 4 focuses on downregulated genes. However, equal numbers of genes are upregulated. Following ACBI1 treatment.

The data showing higher expression of SMARCA4 relative to SMARCA2 in various NB contexts (1B) may not be of great significance as SMARCA4 is expressed at higher levels in most tissues. The smaller differences compared to other cell lines are better illustrated in figure 1A and B.

Dear Editor and Reviewers,

Thank you sincerely for your valuable comments and insightful critiques on our manuscript. Your feedback has significantly enhanced the quality of our work. Below, and on the subsequent pages, we present our detailed responses addressing each of your concerns. These responses are highlighted in blue, while the revised portions in the manuscript are colored in red.

Referee #1:

EMBOJ-2023-115707

General summary and opinion about the principle significance of the study, its questions and findings:

The manuscript titled "Targeting SWI/SNF ATPases to Counteract Neuroblastoma Lineage Plasticity and Chemotherapy Resistance" presents an informative investigation into the role of the SWI/SNF chromatin remodeling complex in Neuroblastoma (NB), providing valuable insights into its implications for tumor cell heterogeneity and therapy responsiveness. Xu et al. effectively demonstrate that targeting SWI/SNF ATPases, particularly utilizing SMARCA2/4 dual degraders, leads to significant inhibition of key cellular processes in NB, such as cell proliferation and invasion, while also exerting a pronounced impact on cellular plasticity. The manuscript highlights these outcomes as central findings with significant therapeutic potential. However, there are concerns regarding the interpretation of the presented experimental results in relation to effects on cell proliferation of the dual degraders. To ensure the manuscript's readiness for potential publication in EMBO Journal, the authors are encouraged to address these concerns by assessing both the proliferative and apoptotic statuses of cells subjected to treatment over a four-day duration with SMARCA2/4 inhibitors. These additional experiments will significantly strengthen the manuscript and contribute to a more robust and comprehensive understanding of the presented findings.

Specific major concerns essential to be addressed to support the conclusions:

- In Figure 2 Xu et.al. present a robust effect on chromatin accessibility and decreased global binding of several transcription factors known to promote an ADRN phenotype in NB cells. It appears that the reduced chromatin accessibility observed with ACBI1 and siSMARCA4 treatments is a global event, which raises the question if not nearly all transcription factors will have reduced average binding to DNA? To address this question authors could add additional CHIP data on other transcription factors that are not affected by SMARCA2/4 inhibition. If nearly all transcription factors have less binding to DNA, it is important to rewrite parts of the discussion and highlight that the observed reduction of DNA binding of core oncogenic transcription factors is not an exclusive event.

We appreciate the reviewer's comments. To investigate whether the reduction in DNA binding of these core TFs following SWI/SNF ATPase depletion is an exclusive event, we conducted ChIP-seq experiments for other proteins, including the cohesin subunit RAD21, the transcription repressor CTCF, and the histone-methylating complex subunit WDR5. Our results revealed decreased average signals for all these proteins at the compacted sites after targeting SWI/SNF ATPases with ACBI1 in IMR32 cells, as demonstrated by metagene plots (updated Fig. S2F). This observation suggests that the observed reduction in DNA binding of core oncogenic TFs is not exclusive; rather, the compacted chromatin resulting from the loss of SWI/SNF ATPases acts as a barrier for protein binding. We have included these

findings in the updated Fig. S2F, the main text on page 9, starting from line 213, and the discussion section on page 20 starting from line 491.

Fig. S2

- In Figure 3 authors show that ACB1 and AU-15330 treatment reduce invasion of NB cells. However, in Figure 1 authors also show that the same concentrations of these inhibitors lead to less cell proliferation. Further experiments are therefore needed to evaluate if impaired invasion is simply because cells are damaged or dying. By monitoring cell proliferation more carefully and staining for apoptotic markers this issue could be addressed. Claiming that GFP expression is indicative of cells being healthy is not sufficient evidence.

As we understand the reviewer’s concern is whether cell death induced by ACB1 or AU-15330 contributes to the decrease in invasion detected in these assays. We initially used GFP to monitor cell death based on the findings cited ([doi.org/10.1002/1097-0320\(20011201\)45:4<237::AID-CYTO10024>3.0.CO;2-J](https://doi.org/10.1002/1097-0320(20011201)45:4<237::AID-CYTO10024>3.0.CO;2-J)), however we agree that orthogonal approaches should be used. Using a 3D spheroid culture condition, we investigated whether apoptosis or cell death contributed to the decreased matrigel cell invasion after SWI/SNF ATPases depletion. We performed apoptosis assays by incubating the cells with Incucyte Caspase-3/7 green dye. However, the cell staining did not yield reproducible results, possibly due to the assay’s incompatibility with 3D cell models. Consequently, we turned to the LDH-Glo Cytotoxicity Assay to measure cell death, which detects LDH released from small numbers of cells in 3D models. The LDH-Glo Cytotoxicity Assay results revealed that treatment with ACB1 and AU-15330 significantly increased cell death in IMR5 and IMR32 cells, but had no effect on BE(2)C cells in this matrigel containing 3D cell culture system (Fig. S3F). Thus, the reduction in matrigel invasion following ACB1 or AU-15330 treatment may be partially attributed to effects on cell death in IMR5 and IMR32 cells, but this is not the case for BE2. We have included this result in the updated Fig. S3F and described it in the main text (pages 12, starting from line 267).

Fig. S3

- Similar to concerns regarding Figure 3, it is not clear how to interpret results in Figure 5. Given that the ACBI1 affects cell proliferation it is not clear whether cellular clones are selected for due to toxicity or if they trans-differentiate. As mentioned in the discussion a barcode model would be preferable to further study these possibilities, however, it would be adequate to provide controls for cell proliferation and cell death in the current experimental setup. Extensive toxicity could indicate that cellular clones are being selected for in this experimental setup.

We appreciate the reviewer's comments. The reviewer suggested investigating whether cellular clones are selected due to toxicity. To explore this, we conducted a cell cytotoxicity assay using the Incucyte Cytotox green dye to evaluate the impact of ACBI1 treatment on SJNBLO12407 cell death. Our observations revealed that ACBI1 treatment induced less than 10% cell death, as indicated by the cells stained with green dye (updated Fig. S5E,F). This cell death was observed in both ADRN-like large spheres and cells outside of the large spheres. Notably, the majority of the cells after ACBI1 treatment are still viable. We described these findings in the main text on page 15 starting from line 365.

During the transition from SCM to FBS-containing media, SJNBLO12407 PDX cells give rise to both ADRN- and MES-like cells. However, the depletion of SWI/SNF ATPases hinders SJNBLO12407 cells' ability to generate as many ADRN- and MES-like cells while promoting neuronal differentiation. These findings are supported by cell morphology changes over time (Movie EV1 and EV2, Fig. 5A, and Fig. S5C), bulk RNA-seq data (Fig. S5K,M), and single-cell RNA-seq results (Fig. 5D,E, Fig. S5N,O). These findings, in combination with the Incucyte cell cytotoxicity assay (Fig. S5D,E), are consistent with a model that the depletion of SWI/SNF ATPases impairs the ability of PDX cells to adopt a heterogenous morphology during the transition from SCM to FBS-containing media, and this impairment may be caused by a combination effect involving the suppression of cell proliferation, induction of cell death, and reduction in the ADRN and MES transcriptional profiles. We discussed these findings in the discussion section (page 22, starting from line 531).

Fig. S5

Minor concerns that should be addressed:

- In Figure 1d what is the rationale to use retinal pigment epithelia (RPE) cell line as a control for neuroblastoma cell lines? Also, does ACBI1 lead to SMARCA2/SMARCA4 degradation in ARPE19?

Several 'normal' cell lines, including HEK 293, exhibit resistance to the SMARCA2/SMARCA4 degrader AU-15330 (<https://doi.org/10.1038/s41586-021-04246-z>). In this study, we chose ARPE-19 as a 'non-tumorigenic' cell line control since it is commonly used in the neuroblastoma research field (DOI: [10.3389/fped.2013.00006](https://doi.org/10.3389/fped.2013.00006); <https://doi.org/10.1038/s41467-022-31331-2>). However, it has not been previously tested for sensitivity to the SMARCA2/SMARCA4 degrader. We have included this information in the main text (page 6, starting from line 121). Our western blot results indicate that both ACBI1 and AU-15330 induce SMARCA2/SMARCA4 degradation in ARPE-19 (Fig. S1L,M), while cell proliferation of ARPE-19 remains unaffected by ACBI1 and AU-15330 treatment.

Fig. S1L, bottom panel

Fig. S1M, bottom panel

- In Figure 1e it is not clear that the degradation is time-dependent. More time points should be added, and quantifications of the western blot bands are required to draw that conclusion.

We thank the reviewer's comments. We performed new western blot assays by including more time points (1, 2, 4, 8, and 24 h) for IMR32 cells treated with various doses of either ACBI1 or AU-15330. Additionally, we included more time points (1, 2, 4, 8, and 24 h) for IMR5 and BE2C cells treated with various doses of ACBI1 for western blotting. Densitometry analysis of the western blot results consistently showed that the degradation of SMARCA2/SMARCA4 after ACBI1 or AU-15330 treatment in NB cell lines is time- and dose-dependent. We included these findings in the updated Fig. S1L,M.

Fig. S1

L NB cell lines treated with ACBI1 at multiple time points

M NB cell lines treated with AU-15330

- In Figure 3 ACBI1 is used at a range of different concentrations from 100nM to 1000nM. Is this a typo? If not, why do different cells require such different concentrations - the amount of SMARCA4 in these cells (Fig 1d) is rather similar.

Although the amount of SMARCA4 is similar in tested NB cell lines (Fig. S1D), both western blot results and cell proliferation assays indicated varying drug sensitivity among different NB cell lines. For instance, BE(2)C cells exhibit the highest resistance to ACBI1 or AU-15330 treatment in two-dimensional cell proliferation assay (Fig. 1E,F), necessitating higher doses to achieve substantial degradation (>80%) of SMARCA2/4 (Fig. S1L,M). In contrast, IMR5 is most sensitive to the degraders, requiring lower doses to achieve both SMARCA2/4 degradation and cell proliferation suppression (Fig. 1E,F, Fig. S1L,M). Therefore, we adjusted our treatment strategy accordingly, for example, using 1000 nM ACBI1 for BE(2)C and 200 nM ACBI1 for IMR5 and IMR32 cells in Fig. 3. The distinct responses of NB cell lines to drug treatment may arise from genetic differences such as p53 mutation which occurs in BE(2)C but not IMR5 or IMR32 (as detailed in the Materials and Methods section). We cannot rule out that differences in the proteasome machinery (doi: [10.1016/j.isci.2022.103985](https://doi.org/10.1016/j.isci.2022.103985); [10.1021/acscchembio.9b00525](https://doi.org/10.1021/acscchembio.9b00525)) in NB

cells may affect their degradation ability, as well as other factors affecting drug penetration and efflux regulators (doi: 10.1016/j.drug.2019.100671). We have included this in the main text (page 7, starting from line 153).

- Similarly in Figure 3d, AU-15330 (200nM or 2000nM) does not seem consistent in the quantified result and image.

We appreciate the reviewer for bringing this to our attention. In the previous Fig. 3D, the top middle panel (the graph showing invading cells brightfield object of IMR5 cells) contained an incorrect graph, mistakenly copied the one from BE(2)C cells treated with ACBI1 and AU-15330 (bottom middle graph). We have now replaced this graph with the correct one in the new Fig. 3D.

- In Figure 3, in the 3D spheroid cell invasion assay, cells were only treated cell once. What is the half-life of ACBI1/AU-15330?

In the 3D spheroid cell invasion assay (Fig. 3), cells received a single treatment of ACBI1, while AU-15330 was administered every 48 hours (page 27, line 665). We have provided clarification on this matter in the Materials and Methods section. We lack the technique to directly measure the half-life of ACBI1/AU-15330. To assess the duration of effectiveness of ACBI1 and AU-15330 drugs in media, we exposed IMR32 cells to varying doses of these compounds for up to 6 days without replenishing the drug supply. Through western blot analysis, we observed that treatment with ACBI1 or AU-15330 led to the degradation of SMARCA2/4, with no subsequent recovery of protein levels, even by day 5 (Fig. S1N,O). Based on this indirect measurement, it appears that the half-life of ACBI1 and AU-15330 exceeds 5 days when used in 2D cell culture conditions. The western blot results have been incorporated into the revised Fig. S1N,O and described in the main text on page 7 starting from line 143, as well as in the Materials and Methods section, page 25, starting from line 629.

Fig. S1

- In Figure 5a there is no visual difference (to the human eye) between the DMSO and ACBI1 conditions and authors should present quantifications of cellular morphologies in main Figure 5.

We appreciate the reviewer's comments. In Fig. 5A and the previous supplementary Fig. 5, we used cropped images that only show a small number of cells to demonstrate the difference between PDX cells treated with DMSO (vehicle control) and those treated with ACBI1. We agree with the reviewer that the visual difference was not very convincing in the photos. To address this, we quantified the cell confluence based on the cell morphologies by using the Incucyte live cell image software, which revealed that ACBI1 treatment suppressed cell expansion (updated Fig. 5B). Additionally, we included uncropped images at different time points in the updated Fig. S5C to display more cells in each image. In the DMSO-treated group, there was a substantial increase in the number of flattened, enlarged, MES-type morphology monolayer cells over time after switching the media from SCM to FBS-containing media (Fig. S5C, Movie EV1). However, ACBI1 treatment impeded the expansion of flattened, enlarged, monolayer cells over time. This effect is more clearly visible in the updated uncropped cell images (Fig. S5C) and the associated Incucyte movies (Movie EV1 and EV2). We have described these updates in the main text, on page 15, starting from line 355.

Fig. 5

Fig. S5C

We attempted to use the Incucyte cell-by-cell analysis tool to quantify individual cell populations (such as spheres and flattened monolayer cells indicated by yellow and red arrows in Fig. 5A), unfortunately, the software couldn't mask each cell population very well in this setting, possibly due to the low contrast of the flat cells. Nevertheless, by quantifying all cells in the DMSO group and ACBI1 group, including uncropped images and movies, it is evident that the depletion of SWI/SNF ATPases impedes the expansion of MES-like flattened monolayer cells.

- In Figure 6d quantifications of cell morphology are needed and statistical analysis to determine differences between treatments.

As recommended, we quantified cell confluences based on the cell morphologies by using the Incucyte live cell image analysis software. The results revealed that single treatments of 250 nM ACBI1 or 500 nM etoposide were significantly less effective than the combination treatment (updated Fig. 6C). As anticipated, the enlarged representative cell images (Fig. 6E) and uncropped cell images (updated Fig. S6B) demonstrated that ACBI1-treated cells predominantly exhibit spheres and neuroblast-like cells, with fewer flattened monolayer cells. In contrast, when exposed to etoposide alone, there was a reduction in ADRN-type cell numbers, but a substantial population of flattened monolayer cells persisted in the cultures (Fig. 6E, updated Fig. S6B). Importantly, the ACBI1 and etoposide combination effectively depleted cells with MES-type morphology and hindered the growth of ADRN-type cells (Fig. 6E, updated Fig. S6B).

Fig. 6

Fig. S6

B SJNBL012407_X1

- In Figure 6g, something looks strange with the SY5Y graph, since the assays were started with similar amounts of cells according to the materials and methods.

We appreciate the reviewer for bringing this to our attention. In the “Materials and Methods” section, we initially specified the number of cells plated in the 96-well plate for 2D cell proliferation when evaluating the activity of ACBI1 and AU-15330 (Fig. 1E, F). However, in Fig. 6F-H, where we investigated the combination effect of ACBI1 and etoposide, we required the use of various doses of each drug. Achieving this in 96-well plates posed challenges, so we opted for 384-well plates instead. We apologize for missing this information initially, and we have now included the cell numbers used for this study with the 384-well plates in the “Materials and Methods” section (page 25, starting from line 613). Specifically, we employed a higher number of SY5Y cells, as we observed that a lower cell count did not proliferate well in the 384-well plate environment.

- The authors should label key oncogene mutation status in the NB cell lines they are using. For example, SH-SY5Y harbors an SMARCA4-R973W mutation. (https://depmap.org/portal/cell_line/ACH-001188?tab=overview)

We appreciate the reviewer's advice. By exploring depmap portal and literature, we have integrated key oncogene mutation status in NB cell lines. This information is listed below and included in the Materials and Methods section on page 24.

SH-SY5Y: SMARCA4, R973W; ALK, F1174L (<https://doi.org/10.1038/nature07397>; depmap portal)

SK-N-SH: SMARCA4, R973W; ALK, F1174L (depmap portal)

SK-N-BE(2)C: TP53, C135F (<https://doi.org/10.1158/0008-5472.CAN-06-0792>); NF1, loss of heterozygosity (<https://doi.org/10.1016/j.cell.2010.06.004>); MYCN amplification.

SMS-KCNR: ALK, R1275Q (<https://doi.org/10.1038/nature07397>); MYCN amplification.

IMR32: NF1, R69G; CCND2, L214I (depmap portal); MYCN amplification.

IMR5: MTOR, F1888V; CCND2, L214I (depmap portal); MYCN amplification.

- In ref 46, Jubierre et al. used the same cell line (BE2(C)) and found knockdown of SMARCA4 induced caspase-dependent apoptosis. Do the authors observe apoptosis in their system? (46. Jubierre, L. et al. BRG1/SMARCA4 is essential for neuroblastoma cell viability through modulation of cell death and survival pathways. *Oncogene* 35, 5179-90 (2016).

To investigate the impact of ACBI1 and AU-15330 on NB cell apoptosis, we treated IMR5, IMR32, and BE(2)C cell lines with varying doses of ACBI1 or AU-15330. Next, we conducted an apoptosis assay by incubating the cells with Incucyte Caspase-3/7 green dye. Representative images of IMR5 cells revealed that approximately 50% of cells treated with 200 nM ACBI or AU-15330 exhibited green fluorescence (Fig. S1Q). Furthermore, Incucyte apoptotic IMR5 cell counts, based on Caspse-3/7 green dye staining, showed a significant increase in cell apoptosis with various doses of ACBI1 or AU-15330 treatment compared to control (Fig. S1Q). However in IMR32 cells and BE(2)C cells, less than 10% of cells showed evidence of apoptosis (green fluorescence) after ACBI or AU-15330 treatment (Fig. S1R,S). These findings indicate that the depletion of SWI/SNF ATPases through PROTAC degraders induces NB cell apoptosis, but the effect varies depending on the specific cell lines. We have included these findings in the updated Fig. S1Q-S, and the main text, on page 8, starting from line 167.

Fig. S1

- Figure 1, the data from DepMap portal could be updated to 23Q2 which includes more NB cell lines. DepMap Public 23Q2+Score, Chronos (n=33) and Expression Public 23Q2 (n=37)

We appreciate the reviewer's advice. We have updated the DepMap data in Fig. 1 to the latest version (23Q4).

Referee #2:

Neuroblastoma disproportionately accounts for 15% cancer related deaths in children. With intensive multimodal therapies, only 50% high-risk patients get cured. While a number of reasons could contribute to the failure of current therapies, it is believed that cellular plasticity of neuroblastoma may play an important role in chemoresistance. It has been long noticed that neuroblastoma cells are a heterogenous population by Dr. Thiele, the senior author of this paper. Recent studies by using RNA-seq and epigenetic approaches confirmed the heterogeneity of neuroblastoma cells at molecular levels, leading to a definition of ADRN and MES types of neuroblastoma cells. While the literature proposed

that ADRN and MES are interconvertible, it remains puzzling that there are not much solid evidence showing the MES neuroblastoma cells are pre-existing in tumors, which are believed to be responsible for chemoresistance. In this elegant study, Drs. Thiele and Liu group made a further contribution by showing that SWI/SNF ATPases are essential in establishing a MES-permissive chromatin state in ADRN neuroblastoma, promoting cell lineage plasticity. By applying ATAC-seq, ChIP-seq and RNA-seq in combination with genetic and pharmacologic degradation of SMARCA4 and SMARCA2, they clearly showed that SWI/SNF ATPases regulate the chromatin accessibility for both ADRN and MES genes in the so called ADRN cell lines. Importantly, they found that SWI/SNF ATPases are important to tumor cell invasion, and the PROTAC degrader against SMARCA4/SMARCA2 suppressed both cell invasion and cell proliferation. Further, they demonstrated that the SMARCA4/SMARCA2 degrader significantly enhanced the anticancer effect of etoposide. Their findings not only provide an insight into understanding the mechanism of neuroblastoma cell plasticity but also a good rationale to target SWI/SNF ATPases to overcome the chemoresistance conferred by MES state. Overall, I really enjoyed reading this MS and recommend publication in EMBO J after clarification.

I have some comments for the authors to consider clarifying .

1. The authors should not overlook the importance of SMARCA2. The DepMap results clearly show that high levels of SMARCA2 is negatively correlated the knockout effect of SMARCA4 (Predictability function in DepMAP after searching SMARCA4). I would suggest the authors add this data into their manuscript, and change tone to address the potential effect of SMARCA2. Also, the presence of SMARCA2 will certainly affect the PROTAC effect against SMARCA4 by titrating the drug. The authors should acknowledge this. The current PROTAC targeting both is not a bad idea at all.

We appreciate the reviewer's comments. By exploring the Predictability function in DepMap, we observed that low levels of SMARCA2 are positively correlated with the CRISPR knockout effect of SMARCA4 when analyzing all the cancer cell lines (Pearson $r=0.333$, $p=7.07E-28$, Fig. S1G, left panel). In NB cell lines, this correlation is not as significant, but there is a trend of positive correlation (Pearson $r=0.278$, $p=1.53E-1$, Fig. S1G, right panel). Additionally, we observed that low levels of SMARCA4 are positively correlated with the CRISPR knockout effect of SMARCA2 in all cancer cell lines (Pearson r value = 0.434 , $p=4.7E-48$, Fig. S1H, left panel), but this correlation is not significant when considering only NB cell lines (Pearson $r=0.106$, $p=5.93E-1$, Fig. S1H, right panel). Nevertheless, NB cell lines with lower expression of SMARCA2 tend to be more sensitive to the loss of SMARCA4, based on this DepMap analysis. These findings suggest that targeting both SMARCA2 and SMARCA4 with PROTAC degraders is a promising approach for treating NB patients. We have described them in the main text (page 6, starting from line 127), and the Discussion section (page 23, starting from line 563).

Fig. S1

2. The authors claimed APRE-19 cell as a normal cell. Actually, no cells are normal once they become immortalized. "Non-cancerous" cell might be more appropriate. It would be helpful for readers to understand the cell of origin of APRE-19 and why the authors chose this cell line as a control.

We agree with the reviewer's comments that ARPE-19 is a 'non-cancerous' cell line. Several 'normal' cell lines, including HEK 293, exhibit resistance to the SMARCA2/SMARCA4 degrader AU-15330 (<https://doi.org/10.1038/s41586-021-04246-z>). In this study, we chose ARPE-19, a human retinal pigmented epithelial cell line as a 'non-cancerous' cell line control since it is commonly used in the neuroblastoma research field (DOI: [10.3389/fped.2013.00006](https://doi.org/10.3389/fped.2013.00006); <https://doi.org/10.1038/s41467-022-31331-2>), and it highly expressed SMARCA4. However, it has not been previously tested for sensitivity to the SMARCA2/SMARCA4 degrader. We included this information in the main text where we mentioned the use of the ARPE-19 cell line (page 6, line 121).

3. In their 3D spheroid experiment, different cell lines were treated with different concentrations of drug. Can they justify this? Can they discuss what potential factors led to the differential responses of neuroblastoma cell lines to the drug treatment?

We applied varying doses of ACBI1 and AU-15330 to different cell lines based on drug sensitivity data obtained from the 2D cell proliferation assay (Fig. 1E,F) and the western blot assay (Fig. 1C,D, Fig. S1L,M). For instance, BE(2)C cells exhibit the highest resistance to drug treatment in 2D cell proliferation assays, necessitating higher doses to achieve a greater than 80% degradation of SMARCA2/4. The distinct responses of NB cell lines to drug treatment may arise from genetic differences such as p53 mutation which occurs in BE(2)C but not IMR5 or IMR32 (as detailed in the Materials and Methods section). We cannot rule out that differences in the proteasome machinery (doi: [10.1016/j.isci.2022.103985](https://doi.org/10.1016/j.isci.2022.103985); [10.1021/acscchembio.9b00525](https://doi.org/10.1021/acscchembio.9b00525)) in NB cells may affect their degradation ability, as well as other factors affecting drug penetration and efflux regulators (doi: [10.1016/j.drug.2019.100671](https://doi.org/10.1016/j.drug.2019.100671)). Consequently, we opted for higher doses to treat more resistant cell lines while selecting lower doses for more sensitive ones in the 3D cell proliferation assay. We have included these reasonings in the main text (page 7, starting from line 153). Additionally, in KCNR cells, the dose range (from 200 nM to 1000 nM) exhibited similar efficacy in degrading SMARCA2/4 (Fig. S1L,M), and we found 250 nM and 500 nM doses of ACBI1 and AU-15330 treatment of KCNR cells in 3D context showed a similar reduction in cell proliferation when compared to 1000 nM drug treatment (Fig. 1G and updated Fig. S1P).

4. In Fig. 2f the authors argued that the effect ChIP-seq signals of H3K27ac and other TFs by siSMARCA4 less than the ACBI1 treatment may be due to lower effect of knockdown, which might be true; however, the authors should discuss the compensatory role of SMARCA2 in IMR32 cells since the western blot showed high levels of expression of SMARCA2.

We agree with the reviewer's comments that the upregulation of SMARCA2 after SMARCA4 knockdown (Fig. S1I) may play a compensatory role that affects the ChIP-seq signals of H3K27ac and other TFs. We have discussed this in the main text on page 9, starting from line 210.

5. In Figure 2h, GSEA results showed a bimodal pattern. Can the authors explain why?

As pointed out by the reviewer, in Fig. 2h, the GSEA results of SWI/SNF ATPases depletion exhibit a bimodal pattern. This occurs because, despite more ADRN signature genes being down-regulated, there are also some ADRN signature genes that are up-regulated (as indicated in Dataset EV3). However, the

GSEA statistical analysis results demonstrate that the depletion of SWI/SNF ATPases leads to a significant negative enrichment of ADRN signature genes. This is primarily due to more ADRN genes being down-regulated rather than up-regulated. We have addressed this phenomenon in the main text (page 11, starting from line 240) and included the list of genes in Dataset EV3.

6. It is very interesting that bulk-RNA-seq identified SCP signature in PDX models when cultured in SCM but shift to ADRN/MES under FBX media. Based on recent scRNA-seq studies to understand the cell of origin of neuroblastoma cells, can author give a further discussion how their findings are linked to the neuroblastoma origin? In Fig. 5c,d, ACBI treatment of FBS cells pushed the ADRN/MES score to a similar degree as in SCM. Does this mean inhibition of SMARCA 4/2 in neuroblastoma cells lead to a more stem cell like state (although not necessarily mean it will increase therapy resistance)? Have they looked into the effect of ACBI1 on PDX cells cultured in SCM?

We thank the reviewer for pointing this out. The analysis of single-cell data derived from patients indicates that a specific subpopulation of tumor cells found in NB map to different stages of SCP/Schwann cell development ([10.15252/emj.2021108780](https://doi.org/10.15252/emj.2021108780); [10.1038/s41588-021-00818-x](https://doi.org/10.1038/s41588-021-00818-x); [10.1073/pnas.2022350118](https://doi.org/10.1073/pnas.2022350118); [10.1101/2020.05.04.077057](https://doi.org/10.1101/2020.05.04.077057)). Our bulk RNA seq analysis of the SJNBL012407_X1 PDX line reveals that when cultured in SCM, it exhibits features similar to SCPs. This observation suggests that the cell origin of SJNBL012407_X1 might be SCPs, or some SJNBL012407_X1 cells reactivate the developmental gene expression signatures, making them similar to SCPs when cultured in SCM. Furthermore, when these cells are cultured in SCM, akin to SCPs, they exhibit greater plasticity and have the potential to give rise to both ADRN and MES cell types upon switching to FBS-containing media. We have discussed this phenomenon on page 21, starting from line 516.

Fig. S5

In the original Fig. 5C,D (now updated Fig. 5D,E), ACBI1 treatment of SJNBL012407_X1 cells cultured in FBS-containing media resulted in ADRN and MES scores comparable to the cells cultured in SCM. However, bulk-RNA seq revealed a positive enrichment of genes associated with neuronal

differentiation following ACBI1 treatment (Fig. S5M). We further analyzed the scRNA-seq data by focusing on the up-regulated neuronal differentiation signatures identified in ACBI1-treated cells based on bulk RNA-seq (Dataset EV8). As indicated in the UMAP plots, more SJNBL012407_X1 cells exhibited high HDA signature (genes expressed in immature dopaminergic neurons) (Fig. S5N) and neuronal differentiation signature scores (Fig. S5O) in ACBI-treated cells cultured in FBS-containing media compared to DMSO-treated cells cultured in FBS-containing media or cells cultured in SCM. These results indicate that, while ACBI treatment of SJNBL012407_X1 cells cultured in FBS-containing results in ADRN and MES scores are similar to those of SCM-cultured cells, the depletion of SWI/SNF ATPases increases the population of cells undergoing neuronal differentiation and decreases cell proliferation. we have included these results in the updated Fig. S5N,O, and discussed these discoveries in the main text on page 17, starting from line 408.

We haven't investigated the effect of ACBI1 on PDX cells cultured in SCM previously. Here we treated SJNBL012407_X1 cells with ACBI1 when the cells were cultured in SCM and found that this treatment also significantly suppressed cell proliferation (Fig. S5D), indicating that SWI/SNF ATPases are essential for PDX cells to proliferate either cultured in SCM or FBS-containing media.

Fig. S5

7. In their scRNA-seq analysis, could the authors give details about the features of each cluster with or without ACBI1 treatment?

We have included additional supplementary tables (Dataset EV6,7) and provided details about the features of each cluster in the scRNA-seq data derived from NB cells cultured in SCM, as well as in FBS-containing media with or without ACBI1 treatment (page 16, starting from line 390). The integrative analysis of all 10,488 cells from the three groups revealed fourteen distinct cell clusters, each characterized by different feature genes (Dataset EV6). While all these clusters were observable in FBS-cultured cells, only six of them (clusters 0, 1, 2, 3, 4, and 6) were detectable in SCM-cultured cells (Fig. 5C, Dataset EV7). This suggests that the remaining seven clusters may consist of cells uniquely induced by the FBS culturing condition. Notably, ACBI1 treatment did not impact the unsupervised clustering of cells in FBS-containing media. Consequently, we focused on the altered signatures identified through bulk-RNA seq for further scRNA-seq data analysis. Specifically, we compared ADRN, MES, and neuronal differentiation signature scores in each cell cultured in SCM, FBS-containing media (with or without ACBI1 treatment). This approach allowed us to clearly observe the effect of ACBI1 treatment on changes in gene expression across multiple clusters.

8. Lastly, the authors should acknowledge that the phenotype they observed was through genetic

depletion and PROTAC-mediated protein degradation. Whether the ATPase activity of SMARCA2/4 is critical to what they observed remains to be studied.

We appreciate the reviewer pointing this out. SMARCA4 and SMARCA2 ATPases are the catalytic subunits of the SWI/SNF chromatin remodeling complexes. In our study, we investigated the function of SMARCA2/4 ATPases through a genetic silencing approach or PROTAC-mediated protein degradation approach. However, a study (<https://doi.org/10.1093/nar/gkad1081>) published during our revision of this manuscript, showed that the treatment of neuroblastoma cell lines with BRM014, an ATPase inhibitor of SMARCA4 and SMARCA2 suppresses neuroblastoma tumor growth and reduces core TFs binding to DNA with decreased chromatin accessibility, which is consistent with our findings, suggesting that the ATPase activity is critical for these processes. We have included this in the discussion section on page 20, starting from line 494.

9. One caveat of this study is lack of *in vivo* test probably due to the issue of PROTAC.

We appreciate the reviewer's comments and acknowledge that drug availability and cost are limitations. Our focus of this study is to investigate the epigenetic mechanisms in determining NB cell invasiveness and plasticity, but our future studies include investigating the effect of targeting SWI/SNF ATPases *in vivo* with or without the combination of chemotherapeutic drugs.

Referee #3:

This manuscript describes the effects of applying recently developed degraders of SMARCA2 and SMARCA4 proteins to NB cells. The compounds cause growth defects in NB cells. To further investigate the underlying mechanisms changes to chromatin accessibility, transcription factor binding and transcription are measured. Loss of transcription factor binding is correlated with downregulation of genes linked to the invasiveness of NB cells. In addition, ACBI treatment reduces invasiveness. Similarly, genes linked to the adrenal and mesenchymal subtypes of NB are reduced following ACBI1 treatment suggesting a role for the ATPases in epigenetic plasticity. It is also observed that combined ACBI1 treatment results in a synergistic reduction in cell numbers. The findings indicate how NB cells respond to loss of the SMARCA ATPases. This will be of interest to researchers studying NB.

The manuscript also mentions the potential for a therapeutic value of SMARCA4 in NB. SMARCA4 has a net negative effect across all cell lines (depmap), its essential for mouse development, loss of function mutations are associated with diverse cancers, so there is a likelihood of side effects when using SMARCA4 inhibitors therapeutically. Indeed, in a human lung xenograft model, toxicity and body weight loss limited therapeutic effects <https://pubs.acs.org/doi/10.1021/acs.jmedchem.8b01318>. Although, NB cells may be more dependent on SMARCA4 than some other cell types there will still be significant potential for off target effects. In order to justify the therapeutic angle presented in the manuscript more information should be provided to identify contexts in which the benefits are likely to outweigh deleterious off target effects.

We appreciate the reviewer's comments and acknowledge the potential off-target effects of targeting SWI/SNF ATPases. As mentioned by the reviewer in this paper ([10.1021/acs.jmedchem.8b01318](https://pubs.acs.org/doi/10.1021/acs.jmedchem.8b01318)), the dual SMARCA2/4 inhibitors showed certain toxicity. This toxicity could be due to targeting SMARCA2/4 in normal tissues in mice or because of targeting other proteins. Interestingly, unlike the dual SMARCA2/4 inhibitors, prolonged treatments with the SMARCA2/4 PROTAC degrader AU-15330 showed

no evident toxicity in immune-competent mice (<https://doi.org/10.1038/s41586-021-04246-z>). AU-15330 does not affect body weight, major organ weight, or the hematologic system of mice. Moreover, our research and other studies have demonstrated that non-cancerous cell lines exhibit greater resistance to SWI/SNF ATPases PROTAC degraders treatment (Fig. 1E,F, <https://doi.org/10.1038/s41586-021-04246-z>). These findings suggest that the SMARCA2/4 PROTAC degraders are less toxic, although we cannot rule out the potential off-target effects in patients. Notably, in our study, the treatment of NB cells with a combination of SMARCA2/4 degrader and a chemotherapeutic drug showed a synergistic effect. Even a low dose of SMARCA2/4 degrader significantly increased the sensitivity of NB cells to chemotherapeutic drug treatment. Therefore, our results suggest that combination therapy could potentially mitigate the off-target effects of targeting SMARCA2/4 in clinical settings. We have discussed this on pages 22-23, starting from line 548.

The data presented appear to be of high quality. The following minor points were noted.

What proportion of accessible chromatin peaks are unaffected by ACBI1?

We thank the reviewer for pointing this out. In the ATAC-seq data analysis of IMR32 cells, we found that among 129452 peaks, ACBI1 treatment (24 h) resulted in decreased chromatin accessibility in 22,070 peaks (17.05%), with around 1,044 peaks (0.81%) showing increased chromatin accessibility (Fig. 2A, left panel). Additionally, 52,630 peaks (40.66%) showed subtle changes (1.25-2.5-fold), while 53,708 peaks (41.49%) remained unaffected by ACBI1 (within 1.25-fold changes). This observation has been included in the main text (page 8, starting from line 180).

ChIP for various factor is shown in figure 2e and f at sites where ATAC-seq is changing. Are changes at these sites representative of the genome wide distribution of these factors? If these factors are also bound at sites where ATAC does not change the effects may be smaller. Data indicating how ChIP changes at sites where chromatin accessibility does not change should also be shown.

Fig. S2

We appreciate the reviewer's comments. These factors also bind at sites where ATAC-seq signals do not change. To determine whether the reduction in DNA binding of these proteins occurs exclusively at sites with decreased chromatin accessibility, we assessed their binding at locations where ATAC-seq signals remained unchanged following ACBI1 treatment. Our findings revealed a dramatic decrease in average

ChIP-seq signals for HAND2, PHOX2B, and H3K27ac, a dramatic increase in GATA3, MYCN signals, and a subtle increase in RAD21 and CTCF signals, while signals for WDR5 did not change (Fig. S2G). The western blot assay indicated that ACBI1 treatment reduced PHOX2B and HAND2 protein levels by approximately 30%, and reduced H3K27ac levels by 15% (Fig. S2D), which possibly contributes to the dramatic decrease in their ChIP-seq signals at both sites with stable and decreased chromatin accessibility. Interestingly, despite no increase in protein levels detected by western blot (Fig. S2D), ACBI1 treatment increased MYCN and GATA3 ChIP-seq signals at the sites without changes in ATAC-seq signals. We postulate that this effect may be due to the re-localization of MYCN and GATA3 from compacted sites to opening chromatin, although we do not have direct evidence to demonstrate this hypothesis. We have included these findings in updated Fig. S2G, and the main text (page 10, starting from line 218).

The effect on ACBI1 on transcription factor occupancy is not uniform. Stronger effects are observed on HAND2 and K27ac, minor effects on MYCN and GAA3. This should be commented. On.

We observed dramatic decreases in HAND2, PHOX2B, and H3K27ac ChIP-seq signals after ACBI1 treatment, but the effect was less pronounced for MYCN and GATA3. The western blot assay indicated that ACBI1 treatment reduced PHOX2B and HAND2 protein levels by approximately 30%, and decreased H3K27ac levels by 15% (Fig. S2D), while MYCN and GATA3 protein levels remained unchanged. Therefore, the changes in the protein levels may contribute to the dramatic decrease in HAND2, PHOX2B, and H3K27ac ChIP-seq signals. We have included this in the main text (page 10, starting from line 223).

The evidence that SMARCA2 plays a limited role in NB is limited. The combined effect of siSMARCA2/4 is not compared to siSMARCA4 alone.

Fig. S1

We agree with the reviewer's comments. We can't rule out the role of SMARCA2 when SMARCA4 is depleted. We haven't compared the combined effect of siSMARCA2/4 to siSMARCA4 alone. However, as recommended by another reviewer, we have explored the Predictability function in DepMap to investigate how the presence of SMARCA2 affects the CRISPR knockout effect of SMARCA4. We observed that low levels of SMARCA2 are positively correlated with the CRISPR knockout effect of SMARCA4 when analyzing all the cancer cell lines (Pearson $r=0.333$, $p=7.07E-28$, Fig. S1G, left panel). In NB cell lines, this correlation is not as significant, but there is a trend of positive correlation (Pearson $r=0.278$, $p=1.53E-1$, Fig. S1G, right panel). In other words, NB cell lines with lower expression of SMARCA2 tend to be more sensitive to the loss of SMARCA4 based on this DepMap analysis. This observation

suggests that targeting both SMARCA2 and SMARCA4 with PROTAC degraders is a promising approach to treating NB patients. We have included these findings in the supplementary Fig. S1G and discussed this in the Discussion section (page 23, starting from line 563).

The transcriptome analysis in Figure 4 focuses on downregulated genes. However, equal numbers of genes are upregulated. Following ACBI treatment.

To address the question raised by the reviewer, we performed transcriptome analysis, focusing on upregulated genes. GSEA gene ontology analysis revealed that the treatment of IMR5 cells with ACBI1 or genetic silencing of SMARCA4 in IMR5 cells resulted in a positive enrichment of genes encoding ribosomal proteins (Dataset EV4). After the depletion of SWI/SNF ATPases in IMR32 cells, the top-ranked positively enriched gene sets were related to DNA repair complex and lens fiber cell differentiation, among others. However, none of these pathways showed significant enrichment (considered significant at $p < 0.05$, FDR $q < 0.25$) (Dataset EV4). This suggests that the depletion of SWI/SNF ATPases in NB cells may impact protein synthesis, but the effect is cell-type-dependent. We have included this finding in Dataset EV4 and described it in the main text (page 11, starting from line 252).

Notably, when focused on the downregulated genes, we observed negative enrichment of EMT, ADRN, and MES genes, as well as genes encoding collagen-containing extracellular matrix proteins across all tested NB cell lines. These findings support our conclusion that SWI/SNF ATPases depletion reduces NB cell migration, invasion, and plasticity – a common phenomenon in NB.

The data showing higher expression of SMARCA4 relative to SMARCA2 in various NB contexts (1B) may not be of great significance as SMARCA4 is expressed at higher levels in most tissues. The smaller differences compared to other cell lines are better illustrated in figure 1A and B.

Comparing the relative abundance of SMARCA4 and SMARCA2 in NB cells could offer one possible reason why SMARCA2 is not as crucial as SMARCA4 in NB. However, as the reviewer noted, this information is not as critical as the other data presented in Figure 1. Consequently, we have relocated the original Figures 1C and D to the supplementary figure, now represented as Fig. S1C,D.

Dear Dr Liu,

Thank you for submitting a revised version of your manuscript. We find that you have addressed all the additional remarks raised by the referee. There remain only a few mainly editorial points that have to be addressed before I can extend formal acceptance of the manuscript:

1. FUNDING INFO: grant number missing in ms: Z01:ZIA BC 010788
 2. Please add up to 5 keywords.
 3. Please rename the COI to "DISCLOSURE AND COMPETING INTERESTS STATEMENT"
 4. CRediT has replaced the traditional author contributions section because it offers a systematic, machine-readable author contributions format that allows for more effective research assessment. Please remove the Authors Contributions from the manuscript and use the free text boxes beneath each contributing author's name in our online submission system to add specific details on the author's contribution. More information is available in our guide to authors.
 5. DATASET EV LEGENDS: please upload the legends as a separate tab in each Excel file
 6. Please save Source data files in a scheme one figure/folder and then uploaded as .zip files. E.g. all the Source data files for figure 1 need to be saved in a single folder and this needs to be zipped and then uploaded as "SD figure 1.zip" file.
- Synopsis:
Papers published in The EMBO Journal are accompanied online by a 'Synopsis' to enhance discoverability of the manuscript. It consists of A) a short (1-2 sentences) summary of the findings and their significance, B) 3-4 bullet points highlighting key results and C) a synopsis image that is 550x300-600 pixels large (width x height, jpeg or png format). You can either show a model or key data in the synopsis image. Please note that the image size is rather small and that text needs to be readable at the final size. Please send us this information together with the revised manuscript.
7. Please note that the specific URLs for GSE240593, GSE240724, GSE183641 datasets are not provided in the data availability statement."
 8. Figure Legends (main + EV): "1. Please note that the exact p values are not provided in the legends of figures 1a-b; 3a; 5i; 6d.
 9. Please indicate the statistical test used for data analysis in the legends of figures 2b, g-h; 3a; 5i."
 10. Please note that the box plots need to be defined in terms of minima, maxima, centre, bounds of box and whiskers, and percentile in the legends of figures 4c-h.
 11. Please note that information related to n is missing in the legends of figures 5d-e.
 12. Although 'n' is provided, please describe the nature of entity for 'n' in the legends of figures 6c-d, h."
 13. Please note that in figure 5a the scale bar unit should be corrected from μM to μm (both in the figure file).
 14. Appendix figure legends, Dataset and movie legends should be removed from ms file

With best regards,

Cornelius

Cornelius Schneider, PhD
Editor | The EMBO Journal
c.schneider@embojournal.org

- a point-by-point response to the referees' comments, with a detailed description of the changes made (as a word file).
 - a word file of the manuscript text.
 - individual production quality figure files (one file per figure)
 - a complete author checklist, which you can download from our author guidelines (<https://www.embopress.org/page/journal/14602075/authorguide>).
 - Expanded View files (replacing Supplementary Information)
- Please see out instructions to authors
<https://www.embopress.org/page/journal/14602075/authorguide#expandedview>

We realize that it is difficult to revise to a specific deadline. In the interest of protecting the conceptual advance provided by the work, we recommend a revision within 3 months (23rd Sep 2024). Please discuss the revision progress ahead of this time with the editor if you require more time to complete the revisions. Use the link below to submit your revision:

Referee #1:

This reviewer appreciates the efforts that the authors have made to address the points raised. I am happy to recommend publication.

Referee #2:

The authors have adequately addressed my concerns.

Referee #3:

In the revised version of the manuscript the issues raised have been addressed. By addressing points raised by all reviewers the manuscript is substantially improved and suitable for publication.

Responses to the editorial points (highlighted in blue):

1. FUNDING INFO: grant number missing in ms: Z01:ZIA BC 010788

We updated the funding information by including the grant number.

2. Please add up to 5 keywords.

We included five keywords immediately preceding the abstract.

3. Please rename the COI to "DISCLOSURE AND COMPETING INTERESTS STATEMENT"

We renamed the COI to "DISCLOSURE AND COMPETING INTERESTS STATEMENT".

4. CRediT has replaced the traditional author contributions section because it offers a systematic, machine-readable author contributions format that allows for more effective research assessment. Please remove the Authors Contributions from the manuscript and use the free text boxes beneath each contributing author's name in our online submission system to add specific details on the author's contribution. More information is available in our guide to authors.

We removed the authors' contributions from the manuscript and included the authors' contributions using the online submission system.

5. DATASET EV LEGENDS: please upload the legends as a separate tab in each Excel file

We uploaded the legends as a separate tab in each Excel file.

6. Please save Source data files in a scheme one figure/folder and then uploaded as .zip files. E.g. all the Source data files for figure 1 need to be saved in a single folder and this needs to be zipped and then uploaded as "SD figure 1.zip" file.

We saved Source data files in a scheme one figure/folder and then uploaded them as .zip files.

Synopsis:

Papers published in The EMBO Journal are accompanied online by a 'Synopsis' to enhance discoverability of the manuscript. It consists of A) a short (1-2 sentences) summary of the findings and their significance, B) 3-4 bullet points highlighting key results and C) a synopsis image that is 550x300-600 pixels large (width x height, jpeg or png format). You can either show a model or key data in the synopsis image. Please note that the image size is rather small and that text needs to be readable at the final size. Please send us this information together with the revised manuscript.

We have uploaded a synopsis image containing the following summary of findings and bullet points:

Neuroblastoma (NB) consists of two primary subtypes: adrenergic (ADRN) and mesenchymal (MES), which can interconvert. We identify that the SWI/SNF complex regulates neuroblastoma cell plasticity by controlling enhancer binding and chromatin accessibility for lineage-specific transcription factors.

- High levels of SWI/SNF ATPases increase chromatin accessibility and enhance DNA binding of core transcription factors.

- In ADRN-type NB cells, high levels of SWI/SNF ATPases reduce the epigenetic barrier, enabling MES-type transcription factors to access DNA.
- High levels of SWI/SNF ATPases promote NB cell plasticity, thus contributing to intra-tumor heterogeneity.
- Targeting SWI/SNF ATPases reduces cellular plasticity, thereby preventing chemotherapy resistance, highlighting their potential as a therapeutic target for NB.

7. Please note that the specific URLs for GSE240593, GSE240724, GSE183641 datasets are not provided in the data availability statement."

We included the specific URLs for GSE240593, GSE240724, and GSE183641 datasets in the "Data availability" statement, and we will make them publicly available as soon as this manuscript is published.

GEO accession number for ChIP-seq, ATAC-seq and bulk RNA-seq data generated in this study is GSE240593 (<https://www.ncbi.nlm.nih.gov/geo/query/acc.cgi?acc=GSE240593>; token ihybsoahtqhpav). The demultiplexed raw sequencing data and processed count matrices for scRNA-seq data were deposited to GEO under the accession number GSE240724 (<https://www.ncbi.nlm.nih.gov/geo/query/acc.cgi?acc=GSE240724>; token glshkmgevbuzxif). RNA-seq data of *HAND2* and *MYCN* silencing for 72 h in IMR32 can be found under GEO accession number GSE183641 (<https://www.ncbi.nlm.nih.gov/geo/query/acc.cgi?acc=GSE183641>; no token is needed for this dataset).

8. Figure Legends (main + EV): "1. Please note that the exact p values are not provided in the legends of figures 1a-b; 3a; 5i; 6d.

We included the exact p values in the updated Figures 1A-B, 3A, 4I, and 6D.

9. Please indicate the statistical test used for data analysis in the legends of figures 2b, g-h; 3a; 5i."

For Figure 2B, in HOMER motif analysis, the Hypergeometric Test is used to determine the statistical significance of the overlap between a set of observed sequences (e.g., DNA sequences containing motifs) and a set of background sequences. Specifically, it calculates the probability of obtaining a given number of target sequences containing a motif purely by chance, given the total number of sequences and the number of sequences that contain the motif.

For Figure G-H, GSEA uses permutation testing to estimate the significance of the enrichment score. This involves the multiple and random shuffling of gene labels to generate a null distribution of the enrichment scores and compares the observed score to this distribution to compute a p-value. The q-value is an adjusted p-value that accounts for the false discovery rate (FDR). The enrichment score (ES) is the maximum deviation from zero during a walk and reflects how overrepresented the genes in a gene set are at the top or bottom of a ranked list of genes.

For Figure 4I, statistical differences were calculated using a two-sided unpaired Student's *t*-test.

We have included the statistical test used for data analysis in the legends of figures 2b, g-h; 3a; 4i.

10. Please note that the box plots need to be defined in terms of minima, maxima, centre, bounds of box and whiskers, and percentile in the legends of figures 4c-h.

In Figure 4C-H, data are presented as box plots, where the middle solid lines indicate the mean, and the whiskers represent min to max. We have included this information in the figure legends.

11. Please note that information related to n is missing in the legends of figures 5d-e.

We have included the information related to n in Figure 5D, E.

12. Although 'n' is provided, please describe the nature of entity for 'n' in the legends of figures 6c-d, h."

In Figures 6C-D and H, the variable 'n' corresponds to the number of wells in either the 96-well plate or the 384-well plate used for each condition. We have described the nature of entity for 'n' in the figure legends.

13. Please note that in figure 5a the scale bar unit should be corrected from μM to μm (both in the figure file).

We corrected the scale bar unit from μM to μm .

14. Appendix figure legends, Dataset and movie legends should be removed from ms file

We removed appendix Figure legends, Dataset, and Movie legends from the manuscript.

Dear Dr. Liu,

I am pleased to inform you that your manuscript has been accepted for publication in the EMBO Journal.

Yours sincerely,

Cornelius Schneider, PhD
Editor
The EMBO Journal
c.schneider@embojournal.org
